# Allele-specific gene-editing approach for vision loss restoration in *RHO*-associated retinitis pigmentosa

Xiaozhen Liu[1,2], Jing Qiao[3], Ruixuan Jia[1,2], Fan Zhang[3], Xiang Meng[1,2], Yang Li[4], Liping Yang[1,2]*

[1]Department of Ophthalmology, Third Hospital, Peking University, Beijing, China; [2]Beijing Key Laboratory of Restoration of Damaged Ocular Nerve, Peking University Third Hospital,, Beijing, China; [3]Beijing Chinagene Co., LTD, Beijing, China; [4]Beijing Tongren Eye Center, Beijing Tongren Hospital, Capital Medical University, Beijing, China

**Abstract** Mutant *RHO* is the most frequent genetic cause of autosomal dominant retinitis pigmentosa (adRP). Here, we developed an allele-specific gene-editing therapeutic drug to selectively target the human T17M *RHO* mutant allele while leaving the wild-type *RHO* allele intact for the first time. We identified a *Staphylococcus aureus* Cas9 (SaCas9) guide RNA that was highly active and specific to the human T17M *RHO* allele. *In vitro* experiments using HEK293T cells and patient-specific induced pluripotent stem cells (iPSCs) demonstrated active nuclease activity and high specificity. Subretinal delivery of a single adeno-associated virus serotype 2/8 packaging SaCas9 and single guide RNA (sgRNA) to the retinas of the *RHO* humanized mice showed that this therapeutic drug targeted the mutant allele selectively, thereby downregulating the mutant *RHO* mRNA expression. Administration of this therapeutic drug resulted in a long-term (up to 11 months after treatment) improvement of retinal function and preservation of photoreceptors in the heterozygous mutant humanized mice. Our study demonstrated a dose-dependent therapeutic effect *in vivo*. Unwanted off-target effects were not observed at the whole-genome sequencing level. Our study provides strong support for the further development of this effective therapeutic drug to treat *RHO*-T17M-associated adRP, also offers a generalizable framework for developing gene-editing medicine. Furthermore, our success in restoring the vision loss in the suffering *RHO* humanized mice verifies the feasibility of allele-specific CRISPR/Cas9-based medicines for other autosomal dominant inherited retinal dystrophies.

*For correspondence:
alexlipingyang@bjmu.edu.cn

## Editor's evaluation

This work provides a valuable allele-specific gene editing therapeutic approach to selectively target the human RHO-T17M mutation, one of the most frequent genetic causes of autosomal dominant retinitis pigmentosa patients. Overall, the data is solid.

## Introduction

Retinitis pigmentosa (RP [MIM:268000]), a subtype of inherited retinal dystrophies (IRDs), is one of the most common causes of blindness in industrialized countries, with an incidence of 1 in 1000 in North China (*Xu et al., 2006*). However, there are currently no reliable therapies available to cure or delay the disease progression of RP. In 2017, the US Food and Drug Administration approved Luxturna, an adeno-associated virus (AAV)-based gene therapy drug for *RPE65*-associated Leber's Congenital

Amaurosis (*RPE65*-LCA) (*Russell et al., 2017*; *Smalley, 2017*), which marked the start of gene therapy for IRDs (*Lee et al., 2019*).

Mutations in the human rhodopsin (*RHO*) gene are the most prominent genetic causes of autosomal dominant RP (adRP), accounting for about 30% of adRP cases (*Iannaccone et al., 2006*). However, different from the *RPE65*-LCA, which is caused by a loss-of-function mutation, *RHO*-adRP is mainly caused by a gain-of-function or dominant-negative mutation (*Li et al., 2018*; *Mendes et al., 2005*), thereby restricting the application of gene replacement therapy.

Many studies have been conducted regarding the treatment technique for *RHO*-adRP, such as allele-independent gene ablation. This technique includes silencing the expression of both the mutant *RHO* allele and its wild-type (WT) counterpart and simultaneously adding an exogenous WT *RHO* gene beyond the regulation of the retina *in vivo* (*Mitra et al., 2018*; *Cideciyan et al., 2018*; *Tsai et al., 2018*). Although these methods are applicable to *RHO*-adRP, many risks remain with the disruption of the WT allele. Recently, allele-specific methods have been used extensively to treat adRP (*Meng et al., 2020*). With the aim of allele-specific *RHO* gene downregulation at the RNA level using antisense oligonucleotides (ASO), short hairpin RNA, or RNA interference (*Meng et al., 2020*) can be achieved, some of which have been translated into clinical trials. However, repeated treatments, especially for subretinal injections, are bound to hamper therapeutic effectiveness. In comparison, allele-specific mutant DNA targeting approaches, specifically with the CRISPR/Cas9 system (*Mali et al., 2013*; *Ran et al., 2015*), which leads to permanent DNA modification, may be more promising. In fact, the CRISPR/Cas9 system has been used extensively to treat adRP caused by *RHO*-P23H, S334ter, P347S mutations, *etc*. This resulted in the delay in photoreceptor degeneration and rescue of retinal function in animal models (*Latella et al., 2016*; *Li et al., 2018*; *Giannelli et al., 2018*; *Patrizi et al., 2021*; *Bakondi et al., 2016*). However, these proof-of-concept studies are not suitable for direct translation into clinical practice. Therefore, there is currently no approved treatment for adRP.

The P23H mutation is the most common *RHO* mutation in North America, accounting for 10% of adRP cases due to a founder effect, while it has not been reported elsewhere, including Europe and Asia (*Meng et al., 2020*). Data gathered from our previous study (*Liu et al., 2021*) showed that some Chinese adRP patients carried *RHO*-T17M mutation. Several studies and data from the Human Gene Mutation Database (HGMD, http://www.hgmd.cf.ac.uk/ac/index.php) also prove that T17M (*Sheffield, 1991*; *Krebs et al., 2010*; *Rakoczy et al., 2011*) is one of the most common *RHO* mutations.

Previous *in vitro* and *in vivo* studies have shown that if the disease-causing allele has unique protospacer adjacent motif (PAM) sequences that are not present in the WT allele, or when the mutations are within the spacer region of the Cas9/single guide RNA (sgRNA) target site, the disease-causing allele may be discriminated from its WT counterpart. Therefore, the aim of selectively inactivating the mutant allele while leaving the WT ones functionally intact can be achieved (*Burnight et al., 2017*; *Li et al., 2018*; *Giannelli et al., 2018*; *Patrizi et al., 2021*). Non-homologous end joining (NHEJ) is the primary mechanism for double-strand breaks in terminally differentiated rod cells, whereas homologous recombination commonly occurs in mitotic cells such as induced pluripotent stem cells (iPSCs) or HEK293T (293T) (*Chan et al., 2011*). This difference in rod cells from mitotic cells emphasizes the significance and necessity of performing experiments *in vivo*, especially in humanized animal models. Unfortunately, allele-specific CRISPR/Cas9 experiments have not been performed in humanized animal models.

Hence, we developed an allele-specific gene-editing therapy to selectively target the human T17M (c. 50C>T) *RHO* mutant allele while leaving the WT *RHO* allele intact for the first time. Our *in vitro* experiments using 293T cells and patient-specific iPSCs demonstrated high nuclease specificity. Furthermore, a human *RHO*-knock-in (*RHO* humanized) mouse model (*Liu et al., 2022*) generated previously can simulate pathological changes in *RHO*-adRP, the retinal degeneration might be caused by any one of the five variants (T17M, G51D, G114R, R135W, and P171R, *RHO*-5m). Using this mouse model, our *in vivo* experiments demonstrated specific inactivation of the mutant *RHO* allele and a delay in photoreceptor degeneration for up to 11 months after treatment. Our study demonstrated a dose-dependent therapeutic effect *in vivo*. Unwanted off-target effects were not observed. Thus, our study clearly provides strong evidence to support further development of this therapeutic drug for patients with *RHO*-T17M-adRP.

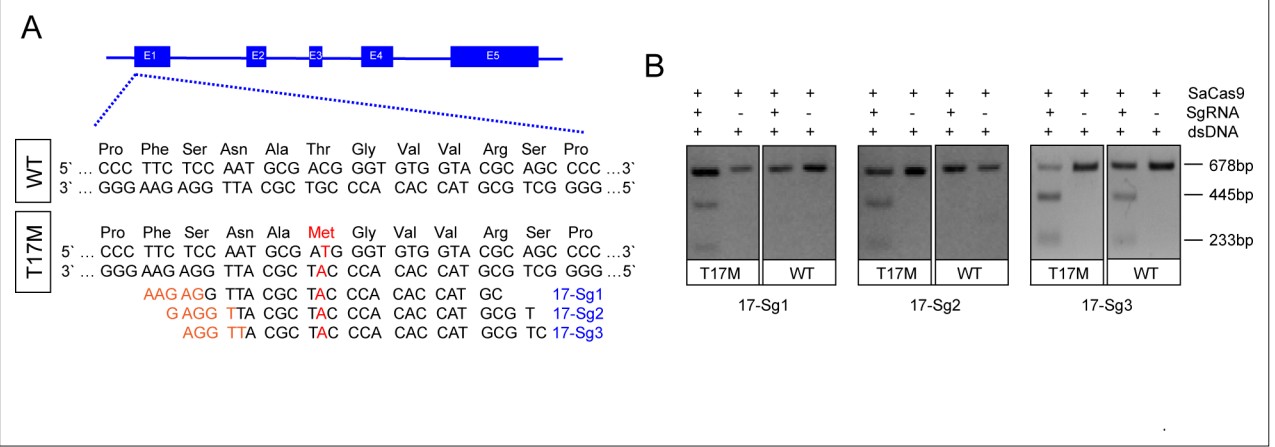

**Figure 1.** Schematic representation of allele-specific sgRNA design for *RHO*-T17M mutation. (**A**) and confirmation of the specificity achieved using SaCas9 protein complexed with an sgRNA to target the mutant *RHO* sequence and its corresponding WT sequence *in vitro* (**B**), the full-length amplicon was 678 bp, the two truncated amplicons were 445 bp and 233 bp, respectively. The mutation c.50C>T (p.T17M) is indicated in red. PAM sequences were marked in orange. Exons were indicated by closed boxes.

## Results

### Identification of allele-specific sgRNAs for the *RHO*-T17M allele

Our objective was to delete the mutant *RHO* allele using an allele-specific sgRNA against mutational sites with NNGRR PAMs when using SaCas9, thereby tethering the SaCas9 protein to the mutant but not the WT allele. The mutant allele, including the T17M mutation, did not generate unique PAM sequences. Allele-specific sgRNAs for T17M were designed to be within the spacer region of the SaCas9/SgRNA target site. For T17M, three sgRNAs, 17-Sg1: 5'-CGTACCACACCCATCGCATTG-3' (PAM: GAGAA), 17-Sg2:5'-TGCGTACCACACCCATCGCAT-3' (PAM: TGGAG), and 17-Sg3: 5'-CTGCGTACCACACCCATCGCA-3' (PAM: TTGGA), were designed to target the mutant allele (*Figure 1A*). *In vitro* digestion of either WT or mutant *RHO* sequence with SaCas9/SgRNA was carried out to assess the specificity profile of these 17-sgRNAs. 17-Sg1 and -Sg2 appeared to preferentially cut the mutant sequence, but not the WT sequence, whereas 17-Sg3 cut both the WT and mutant sequences and appeared to cut the mutant sequence more than the WT sequence (*Figure 1B*).

### *In vitro* knockdown of *RHO*-T17M gene expression by allele-specific gene editing in 293T cells

Due to the unavailability of cell lines expressing human *RHO*, we created 293T cell pool stably expressing *RHO*-WT or *RHO*-T17M using lentivirus FUGW-*RHO*-cDNA (*Figure 2A*). Subsequently, 293T cells expressing *RHO*-WT (RHO*wt* cells) and RHO17 cells were transfected with pX601-EFS-SaCas9-U6-17-Sg1 (17-Sg1) or -Sg2 plasmid (*Figure 2A*). Notably, RHO17 cells that were not transfected assayed with T7E1 nuclease only resulted in full-length amplicon (760 bp, *Figure 2B*). Similarly, T7E1 assay for RHO*wt* cells transfected with 17-Sg1 or -Sg2 plasmid resulted in full-length amplicon (*Figure 2B*). In contrast, the T7E1 assay for RHO17 cells transfected with 17-Sg1 or -Sg2 plasmid demonstrated that 17-Sg1 and -Sg2 were able to specifically cut the mutant sequence (*Figure 2B*). Subsequent TA cloning (59 clones) were chosen for each sgRNA and Sanger sequencing were used to measure the frequency of insertion or deletion mutations (indels). Indels were not detected in WT samples. However, indels were found at frequencies of 11.86% (7/59) and 25.42% (15/59) in samples of SaCas9/17-Sg1 and SaCas9/17-Sg2, respectively (*Figure 2C*, *Supplementary file 1a*). Moreover, we evaluated rhodopsin protein production through western blotting (WB). Rhodopsin expression in RHO17 cells transfected with 17-Sg1 or -Sg2 plasmid was strongly reduced compared to the counterpart of RHO*wt* cells with 17-Sg1 or -Sg2 plasmid (*Figure 2D–E*).

To further analyze the effect of allele-specific knockdown of the *RHO*-T17M, we generated 293T cells that expressed SaCas9/17-Sg1 or SaCas9/17-Sg2 using lentivirus SaCRISPR_SgRNA-GFP. These 293T cells were then transfected with *RHO*-pmCherryN1 plasmids with or without the T17M mutation (*Figure 2—figure supplement 1A*). Seventy-two hours post transfection, fluorescence intensity was

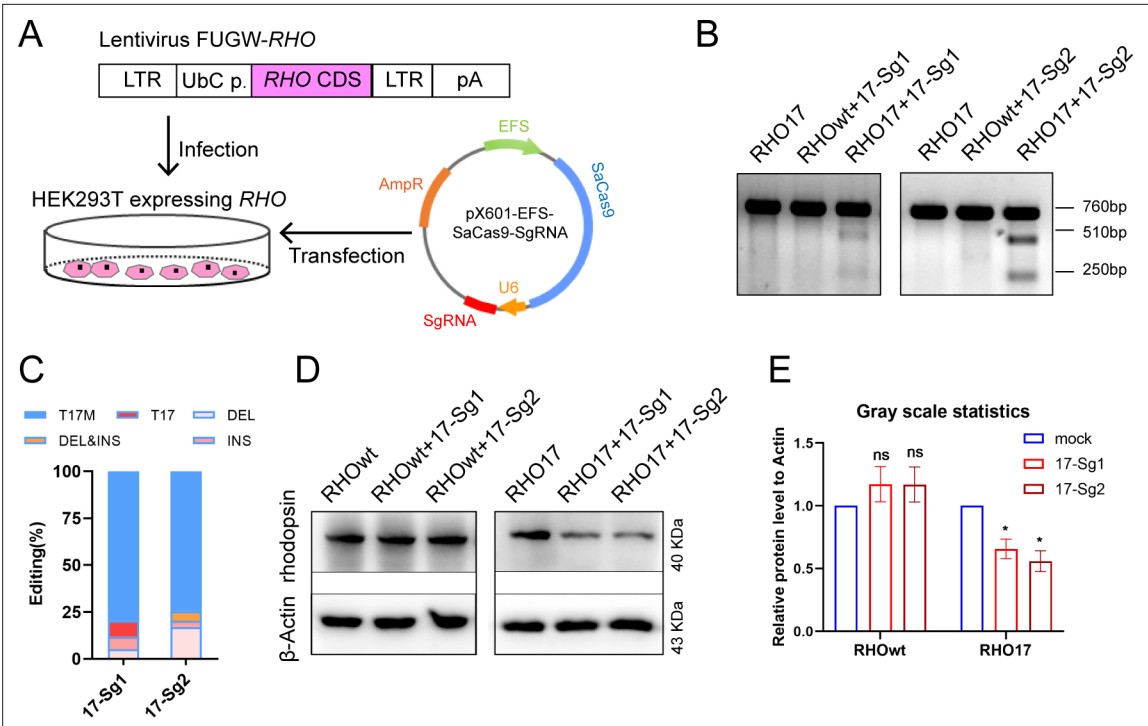

**Figure 2.** *In vitro* knockdown of human *RHO*-T17M expression. (**A**) Schematic view of construction of 293T stably expressing human *RHO* protein and transfection of pX601-EFS-SaCas9-U6-sgRNA (SgRNA) plasmid. (**B**) T7E1 assay indicated that SaCas9/17-Sg1 and SaCas9/17-Sg2 were appeared to cut the mutant sequence specifically, the full-length amplicon was 760 bp, the two truncated amplicons were 510 bp and 250 bp, respectively. (**C**) The cutting efficacy of two sgRNAs with SaCas9 determined by TA and Sanger sequencing in 293T cells. (**D**) Rhodopsin expression reduction was determined by WB in RHO17 cells transfected with 17-Sg1 and -Sg2 plasmid, comparing to the RHO*wt* cells with 17-Sg1 and -Sg2 plasmid. (**E**) Densitometric analysis of immunoblots performed on RHO*wt* and RHO17 cells transfected with 17-Sg1 and -Sg2 plasmid, respectively. The experiment was performed in triplicate and presented as mean ± SEM, the significance was calculated using two-tailed paired *t*-test, ns = not significant, *p<0.05.

The online version of this article includes the following figure supplement(s) for figure 2:

**Figure supplement 1.** The knockdown of human *RHO*-T17M gene expression in 293T cells stably expressing SaCas9 and SgRNA.

examined, and the mCherry/GFP fluorescence intensity ratio (mCherry/GFP ratio) was assessed. There was no significant difference in the mCherry/GFP ratio between 293T cells expressing only SaCas9 (Lenti-CTRL cells) transfected with *RHO*-WT-pmCherryN1 (RHO*wt*) plasmid, and 17-Sg1 or 17-Sg2 cells with RHO*wt* plasmid (*Figure 2—figure supplement 1B–C*). The mCherry/GFP ratio in 17-Sg1 and 17-Sg2 cells transfected with the RHO17 plasmid was significantly reduced compared to that of Lenti-CTRL cells transfected with the RHO17 plasmid (*Figure 2—figure supplement 1D–E*). These results evidently indicated that SaCas9/17-Sg1 and SaCas9/17-Sg2 enabled discrimination between WT and mutant alleles, targeting the mutant allele exclusively. The 17-Sg2 had higher cutting efficiency than 17-Sg1 and was chosen for further *in vitro* and *in vivo* experiments.

## Allele-specific gene editing using CRISPR/Cas9 in patient-derived iPSCs

Urine cells (UCs, *Figure 3A*) from Patient 1 (P1) diagnosed with RP (*Liu et al., 2021*) caused by *RHO*-T17M mutation were obtained and reprogrammed to iPSCs to further determine the specificity and cutting efficiency of SaCas9/17-Sg2 in patient cells. Patient-derived iPSCs manifested a typical cobblestone-like morphology (*Figure 3B*). Sanger sequencing confirmed that patient P1 carried the heterozygous variant c.50C>T (p.T17M) in *RHO* (*Figure 3C–D*). Immunofluorescence (IF) staining showed that their iPSCs expressed human embryonic stem cell-specific surface antigens (*International Stem Cell Initiative et al., 2007*) including NANOG, OCT4, TRA-1–60, and SSEA-4 proteins (*Figure 3E–H*). Chromosomal content analysis revealed a normal 46, XY karyotype (*Figure 3I*). P1 iPSCs (P1 iPS) and iPSCs from a normal donor (NoR iPS) were transfected with the AAV-EFS-SaCas9-U6-17-Sg2-p2a-Puro (17-Sg2-Puro) plasmid (*Figure 3J*). The genomic DNA (gDNA) from these iPSCs

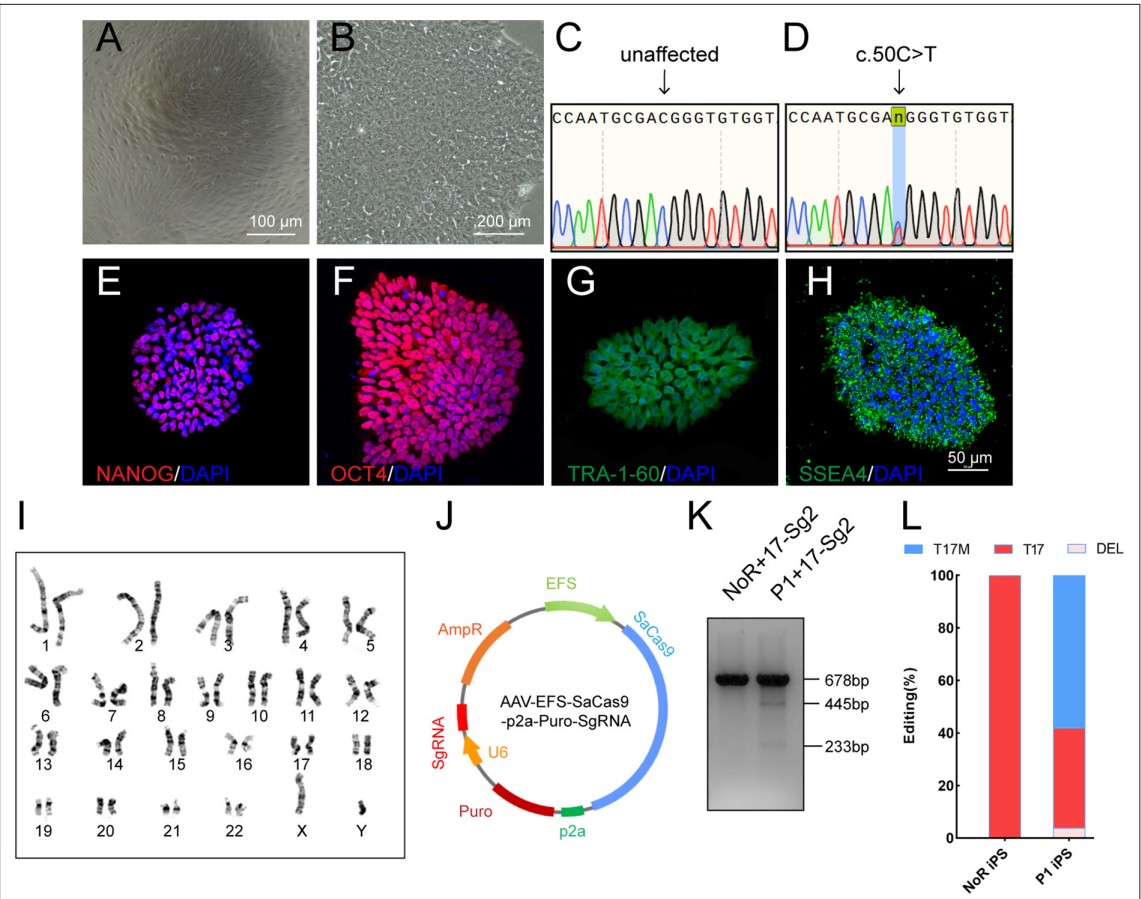

**Figure 3.** Determination of specificity and cutting efficiency of SaCas9 and sgRNA in patient-specific iPSCs. (**A**) Photograph of UCs. Scale bar = 100 μm. (**B**) Photograph of iPSCs. Scale bar = 200 μm. (**C–D**) Sanger sequencing results showed that P1 iPS had the heterozygous c.50C>T mutation. IF staining showed the iPSCs expressed human embryonic stem cell-specific surface antigens including NANOG (**E**), OCT4 (**F**), TRA-1–60 (**G**) and SSEA-4 (**H**) protein. (**I**) Chromosomal content analysis revealed a normal 46, XY karyotype of patient P1. (**J**) Schematic view of AAV-EFS-SaCas9-U6-sgRNA-p2a-Puro plasmid. (**K**) T7E1 assay indicated that SaCas9/17-Sg2 was appeared to cut the mutant sequence specifically. (**L**) The cutting efficacy of SaCas9/17-Sg2 is determined by TA and Sanger sequencing in iPSCs.

The online version of this article includes the following figure supplement(s) for figure 3:

**Figure supplement 1.** Hi-Tom sequence results of SaCas9/17-Sg2-treated P1 iPS (**A**) and P1 iPS colony (**B**).

was subjected to T7E1 assay. The results indicated that 17-Sg2 appeared to specifically cut the mutant sequence (*Figure 3K*). Hi-Tom sequencing revealed that the cutting efficiency of 17-Sg2 was 3.94% (*Figure 3L* and *Figure 3—figure supplement 1A*). Moreover, 17-Sg2 did not facilitate SaCas9-mediated cleavage of the WT allele (*Figure 3K*), and indels were not found (*Figure 3L*) in transfected NoR iPS. To further confirm this allele-specific cutting, colonies from P1 iPS were picked and transfected with the 17-Sg2-Puro plasmid. Similarly, a cutting efficiency of 2.47% was detected (*Figure 3—figure supplement 1B*), a 3bp-deletion (c.44_46del3bp) in the *RHO*-T17M allele was found, whereas no indels were found in the *RHO*-WT allele. These results firmly indicated that the allele-specific 17-Sg2 targeted the mutant allele specifically in patient-derived iPSCs.

### Specific editing of the mutant *RHO* allele in humanized mice

After confirming the ability of SaCas9/17-Sg2 to target the *RHO*-T17M allele specifically in both 293T and patient-derived iPSCs, we aimed to develop gene therapy drug using 17-Sg2 and SaCas9. We chose AAV2/8 as the vector for the drug development because of its superior infectivity to photoreceptor cells. As the first step, we tested if the shortend elongation factor alpha (EFS) promoter was an appropriate promoter to drive strong gene expression in the mouse retina using GFP as a reporter. We constructed AAV2/8-EFS-EGFP (AAV-EGFP) and subretinally injected it into both eyes of the WT

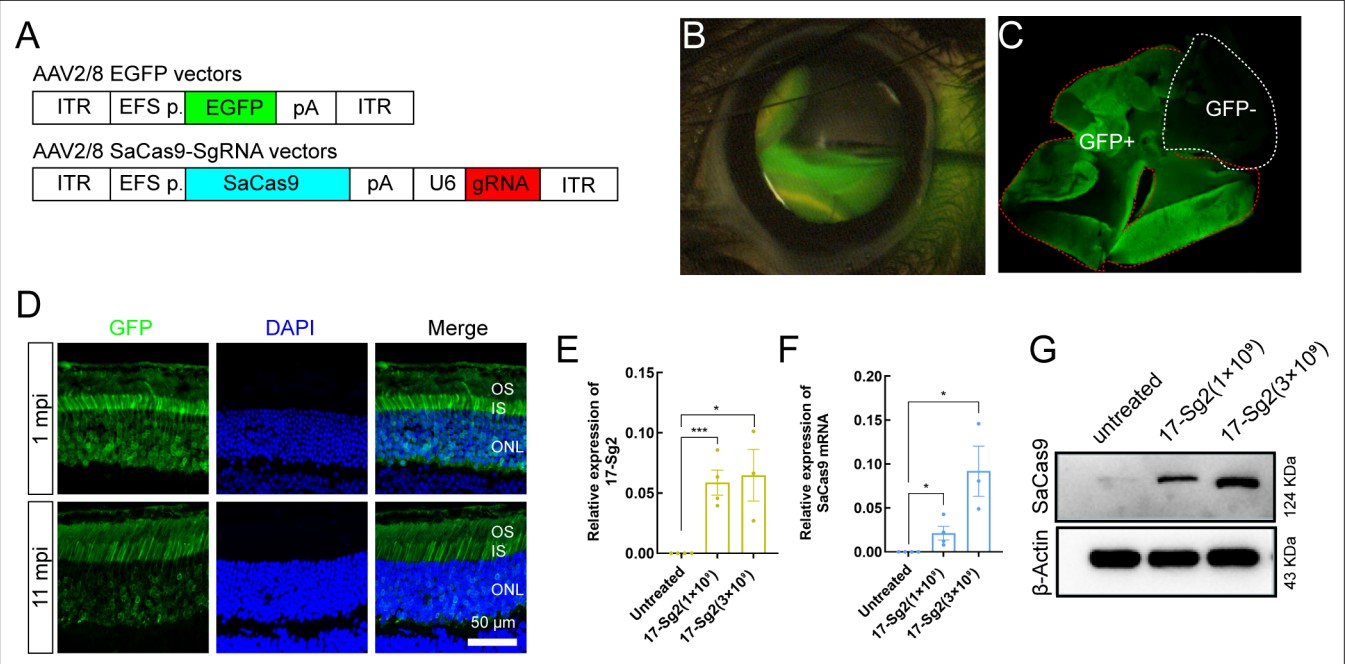

**Figure 4.** AAV2/8-mediated SaCas9 and sgRNA expression in mouse retinas. (**A**) (Top) AAV2/8 vectors expressing EGFP under the control of EFS promoter. (Bottom) AAV2/8 vectors expressing SaCas9 under the control of EFS promoter and sgRNA under the control of U6 promoter were schematized. (**B**) Fluorescein shows AAV vectors distribution following subretinal injection immediately. (**C**) Infected areas treated with AAV-EGFP vector at dose of $1\times10^9$ dose were labeled by GFP expression (GFP+area). (**D**) IF staining indicated that GFP expressed in mouse retinas at 1 mpi and 11 mpi treated with AAV-EGFP vector at $3\times10^9$ dose. ONL = outer nuclear layer; OS = outer segment; IS = inner segment. Scale bar = 50 μm. (**E–F**) QPCR analysis indicated the expression of 17-Sg2 and SaCas9 in $Rho^{wt/hum}$ retinas at 3 mpi treated with AAV-SaCas9/17-Sg2 and AAV-EGFP vector (1:1 mixture) at different doses. Error bars show SEM, and the significance was calculated using two-tailed unpaired $t$-test, *$p<0.05$; ***$p<0.005$. (**G**) Determination of SaCas9 expression by WB analysis at 3 mpi at $1\times10^9$ and $3\times10^9$ dose.

($Rho^{wt/wt}$) mice at either high ($3\times10^9$ vg per eye, 1 μL) or low ($1\times10^9$ vg per eye, 1 μL) dose (*Supplementary file 1b*, *Figure 4A*). Injection blebs were imaged immediately following injection (*Figure 4B*). One month post injection (1 mpi), the neural retinas of the mice were dissected, GFP positive (GFP+) and negative (GFP-) areas were analyzed (*Figure 4C*). IF analysis indicated that GFP was expressed in the outer and inner segment (OS/IS) and outer nuclear layer (ONL) of photoreceptor cells at 1 mpi and 11 mpi (*Figure 4D*). These data indicated that the EFS promoter derived sufficient gene expression in photoreceptors, meeting the requirement for therapeutic drug development. To evaluate whether the EFS promoter could similarly express SaCas9 in the mouse retina, we constructed an expression vector AAV2/8-EFS-SaCas9-U6-17-Sg2 (AAV-SaCas9/17-Sg2) and produced AAVs. To recognizing the subretinal injection sites and considering the limited cargo capacity of AAV, AAV-EGFP and AAV-SaCas9/17-Sg2 (1:1 mixture) were co-delivered into both eyes of the $Rho^{wt/hum}$ (humanized heterozygotes with a mouse $Rho$-WT allele and human $RHO$-WT allele) mice at different doses, each at $5\times10^8$ ($1\times10^9$ vg per eye totally, 1 μL), $1\times10^9$ ($2\times10^9$ vg per eye totally, 1 μL), and $3\times10^9$ ($6\times10^9$ vg per eye totally, 2 μL) dose (*Figure 4A*, *Supplementary file 1b*). As shown in *Figure 4E and F*, quantitative PCR (qPCR) analysis revealed that 17-Sg2 and SaCas9 were expressed in all injected retinas at 3 mpi. WB analysis indicated that SaCas9 was present in injected retinas at 3 mpi (*Figure 4G*) at doses of $1\times10^9$ and $3\times10^9$.

Previously, we generated $RHO$ humanized mouse models using CRISPR/Cas9 and donor plasmids by replacing the mouse $Rho$ gene with the human counterpart (*Liu et al., 2022*). $Rho^{wt/hum}$ mice were heterozygotes with a mouse $Rho$-WT allele and human $RHO$-WT allele, and $Rho^{hum/hum}$ mice were homozygotes with human $RHO$-WT allele. Mut-$Rho^{wt/hum}$ mice were heterozygotes with a mouse $Rho$-WT allele and human $RHO$-5m allele, Mut-$Rho^{hum/hum}$ mice were homozygotes with $RHO$-5m allele. To demonstrate the ability of the allele-specific SaCas9/SgRNA system to edit the human $RHO$-T17M allele *in vivo*, the Mut-$Rho^{wt/hum}$ mouse model was used to test cutting efficiency and $Rho^{wt/hum}$ to evaluate gene-editing specificity (*Supplementary file 1b*). AAV-EGFP and AAV-SaCas9/17-Sg2

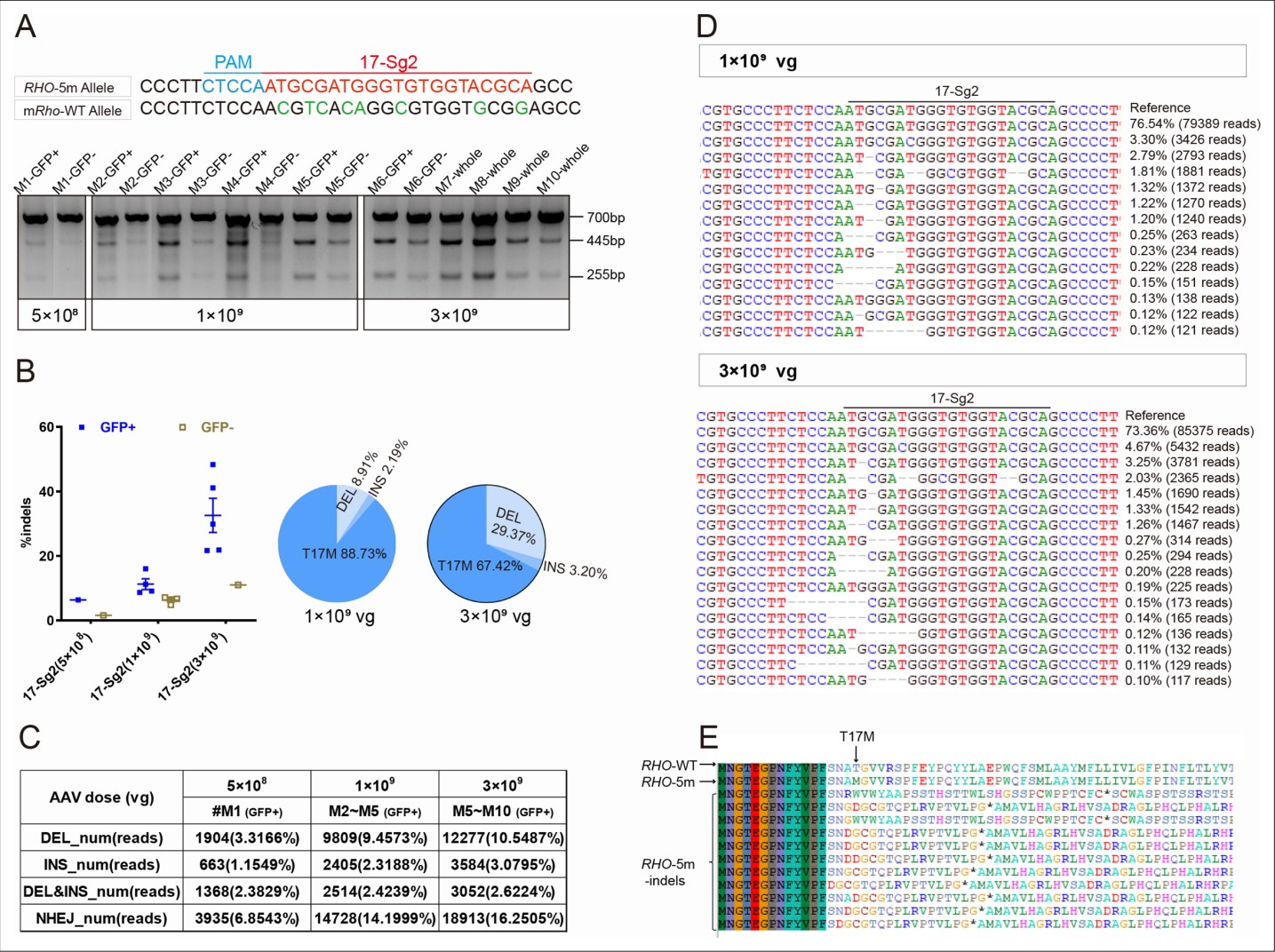

**Figure 5.** CRISPR/Cas9 targeting of c.50C>T *RHO* dominant variant encoding p.T17M in Mut-*Rho^wt/hum* mouse retinas. (**A**) T7E1 assay was performed on GFP+ and GFP- retinas to detect gene-editing activity of AAV-SaCas9/17-Sg2 in Mut-*Rho^wt/hum* mouse retinas at 5×10⁸, 1×10⁹, and 3×10⁹ doses, the full-length amplicon was 700 bp, the two truncated amplicons were 445 bp and 255 bp, respectively; 17-Sg2 was marked in red; PAM sequence was marked in light blue and mismatches in mouse *Rho* sequence comparing to 17-Sg2 were indicated in green. Hi-Tom sequencing (**B**) and PCR-based NGS (**C**) were used to detect the gene-editing efficiency and types of indels in AAV-SaCas9/17-Sg2-treated Mut-*Rho^wt/hum* mouse retinas at 5×10⁸, 1×10⁹, and 3×10⁹ doses. (**D**) Graphic representation of indels scored in the target site of Mut-*Rho^wt/hum* mouse retinas treated with AAV-SaCas9/17-Sg2 at 1×10⁹ and 3×10⁹ doses. (**E**) Analysis of the amino acid sequence of the edited *RHO*-5m allele, * indicated the PTC.

The online version of this article includes the following figure supplement(s) for figure 5:

**Figure supplement 1.** Analysis of gene-editing specificity and efficiency in *RHO* humanized mouse retinas.

**Figure supplement 2.** Phenotypes of the humanized mouse models.

(1:1 mixture) were co-delivered into both eyes of Mut-*Rho^wt/hum* and *Rho^wt/hum* mice, each at 5×10⁸, 1×10⁹, and 3×10⁹ dose. GFP+ and GFP- areas of injected retinas or whole retinas (indicating GFP+ areas could be observed in all retinas) were harvested at 3 mpi. T7E1 assay showed that SaCas9/17-Sg2 was able to cut the mutant sequence specifically (*Figure 5A*, *Figure 5—figure supplement 1A–B*). Truncated cleaved products were detected in Mut-*Rho^wt/hum* mice retinas (mouse #M1–M10) but not in *Rho^wt/hum* mice retinas (mouse #C1–C8) at three different doses. No truncated cleaved products were detected in two Mut-*Rho^wt/hum* mouse retinas (5×10⁸ dose, data not shown). The average efficiencies calculated using TA and Sanger sequencing were 3.33% (GFP+, n=1), 13.99% (GFP+, n=4), 16.59% (GFP+, n=5) in the 5×10⁸, 1×10⁹, and 3×10⁹ dose groups, respectively (*Supplementary file 1c*). Hi-Tom sequencing indicated the average cutting efficiencies of SaCas9/17-Sg2 were 6.37% (GFP+, n=1) and

1.60% (GFP-, n=1), 11.27% (GFP+, n=4) and 6.10% (GFP-, n=4), 32.58% (GFP+, n=5) and 11.04% (GFP-, n=5) at $5\times10^8$, $1\times10^9$, and $3\times10^9$ dose, respectively (*Figure 5B*). PCR-based next-generation sequencing (NGS) indicated that the average efficiencies were 6.85% (GFP+, n=1), 14.20% (GFP+, n=4), 16.25% (GFP+, n=5) in the $5\times10^8$, $1\times10^9$, and $3\times10^9$ dose groups, respectively (*Figure 5C–D*). Therefore, the cutting efficiency showed a dose response. We observed that 89.53% (77/86, TA and Sanger sequencing) of the indels were frameshift (*Supplementary file 1c*), which were expected to create a premature termination codon (PTC) and thereby resulted in the nonsense-mediated decay (NMD) of the edited *RHO*-5m mRNA (https://www.genecascade.org/MutationTaster2021). Among these, c.46_46delG (p. A16Rfs*32), the most common deletion, accounted for 20.93% (18/86) of all indels. We found that the translation of majority of the edited *RHO*-5m protein could be stopped due to PTC (*Figure 5E*).

To determine the off-target activity of 17-Sg2 in the mouse *Rho* allele (8 mismatches between 17-Sg2) and the corresponding mouse *Rho* sequence (*Figure 5—figure supplement 1C*), T7E1 assay was performed and the data clearly showed that SaCas9/17-Sg2 did not cut the mouse *Rho* sequence (*Figure 5—figure supplement 1C*). To provide additional support evidence, AAV-EGFP and AAV-SaCas9/17-Sg2 (1:1 mixture) were similarly co-delivered into the eyes of Mut-*Rho^hum/hum^* mice at $1\times10^9$ and $3\times10^9$ dose (*Supplementary file 1b*). However, the Mut-*Rho^hum/hum^* mice suffered from extremely rapid degeneration of photoreceptor cells and defect formation of OS which was caused by retaining the *RHO*-5m proteins in RIS and cell bodies (*Figure 5—figure supplement 1D*). Moreover, there was no cutting activity detected in the T7E1 assay (*Figure 5—figure supplement 1E*). This result might indicate that infected photoreceptor cells all died at 3 mpi (the age of the mice was approximately postnatal 4.5 month (P4.5m)), photoreceptor cells were terminally differentiated neurons without regeneration capacity, and their existence was essential for the application of gene therapies.

These results from humanized mouse models obviously demonstrated that 17-Sg2 specifically target the *RHO*-T17M allele and might restore the retinal function and vision loss in these mice.

## Decreased expression level of the *RHO*-5m mRNA in AAV-SaCas9/17-Sg2-treated *Rho^hum/m-hum^* retinas

To verify whether the indels created by the activity of SaCas9/17-Sg2 in Mut-*Rho^wt/hum^* mice could lead to attenuation or ablation of the *RHO*-5m allele, we cloned the three most frequent deletions – c.46_46delG (p. A16Rfs*32), c.45_46delTG (p. A16Dfs*17), and c.43_46delAATG (p. N15Rfs*32) and an insertion mutation (c.46_47ins49bp; p. A16Gfs*34) in the edited allele (*Figure 6A*) – overexpressed them as well as *RHO*-WT and *RHO*-T17M in 293T cells. After 1 week of transfection, IF staining using 4D2 antibody (antibody recognizing N-terminal of rhodopsin protein, red) showed that the *RHO*-WT protein was localized on the plasma membrane, while the *RHO*-T17M protein was retained in the cytoplasm in 293T cells (*Figure 6B*). The edited *RHO*-T17M protein of these four indels was undetectable using the 4D2 antibody (*Figure 6B*). Similar results were obtained for the *RHO*-5m variant plasmid. Twenty-four hours after transfection, the number of GFP+ cells per 5000 µm² in 293T cells transfected with different variant plasmids was statistically lower than that in 293T cells with the *RHO*-5m-PEGFPN1 plasmid (*Figure 6—figure supplement 1A–B*). One week after transfection, IF staining using the 4D2 antibody showed that the *RHO*-5m protein was retained in the cytoplasm of 293T cells, while the *RHO*-5m protein of these four indels was undetectable using the 4D2 antibody (*Figure 6C*). GFP+ cells per random sight were counted, and the percentage of GFP+ cells expressing rhodopsin in different variant plasmid groups was calculated (*Figure 6D*), the number of GFP+ cells in the different variant plasmid groups was statistically lower than that in the *RHO*-5m-PEGFPN1 plasmid group, and rhodopsin was rarely expressed in 293T cells transfected with different variant plasmids (*Figure 6E–F*). Taken together, the indels that occurred in the *RHO*-5m allele led to frameshifts, consistent with the results by *in silico* analysis. The *in vitro* results also indicated that although five variants in the mutant *RHO* allele in Mut-*Rho^wt/hum^* mouse, indels created by gene editing around the T17M mutation (variant closest to the N terminal of rhodopsin) could lead to the ablation of the *RHO*-5m mRNA. Thereby, as the five variants are *in cis* on one allele, allele-specific gene editing on T17M alone would erase the effect of other four variants because such editing creates frameshift variants at the N-terminal region before the other four. These data indicated that Mut-*Rho^wt/hum^* mouse could be a practical model which could be used to evaluate phenotype rescues after treatment.

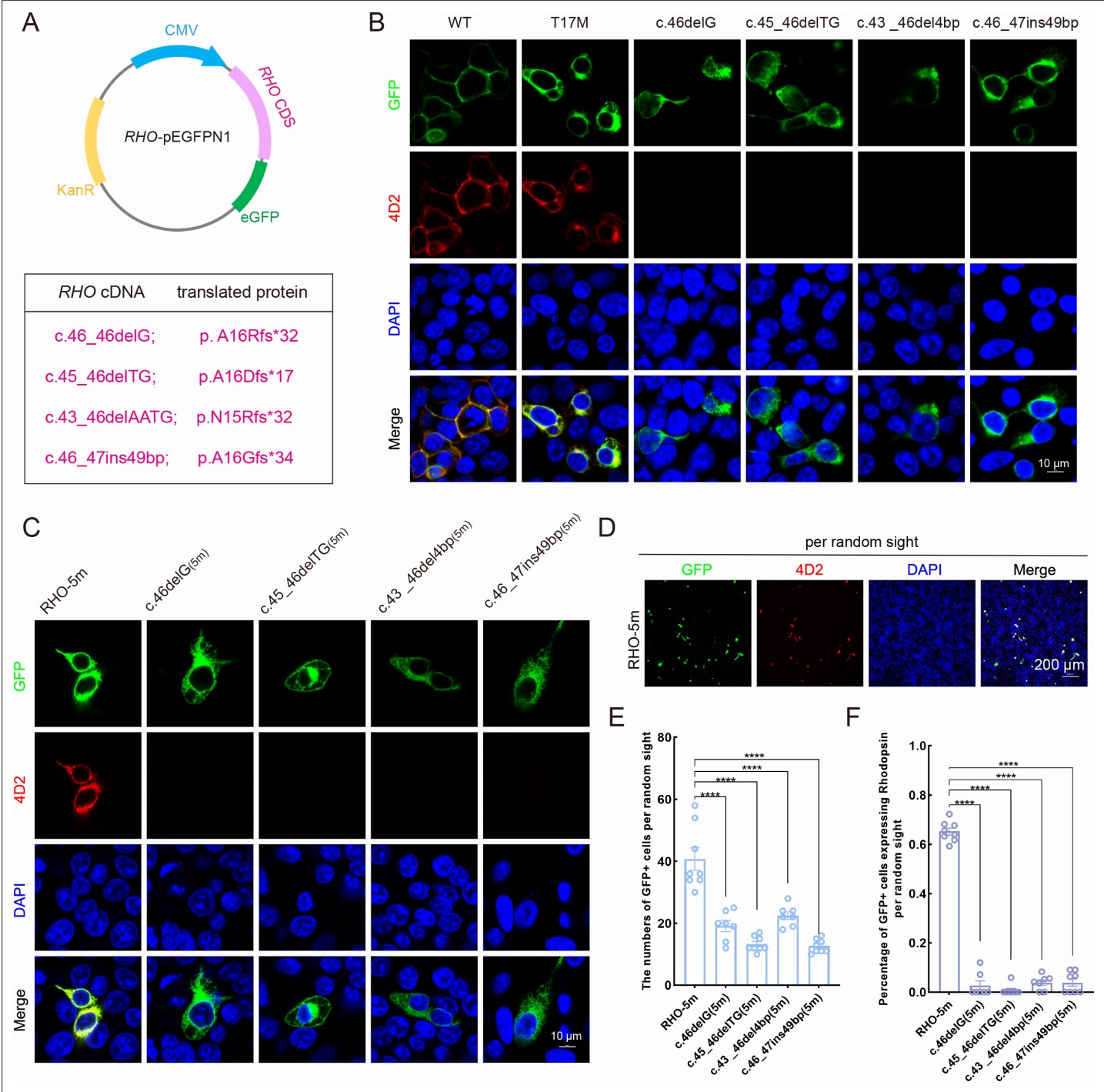

**Figure 6.** Expression of the mutant human *RHO* allele after gene editing with SaCas9/17-Sg2 *in vitro*. (**A**) Schematic view of the different human *RHO* gene variants created by gene editing. (Top) Map of the pEGFPN1 vector used to overexpress these variants. (Bottom) The description of variants at DNA and protein level. (**B**) Colocalization of GFP and rhodopsin (4D2, red) in 293T cells transfected with pEGFPN1 vector carrying *RHO*-WT, *RHO*-T17M, and four edited *RHO*-T17M variants, 1 week after transfection. Scale bar = 10 μm. (**C**) Colocalization of GFP and rhodopsin (4D2, red) in 293T cells transfected with pEGFPN1 vector carrying *RHO*-5m and four edited *RHO*-5m variants, 1 week after transfection. Scale bar = 10 μm. (**D–F**) The number of GFP+ cells and percentage of GFP+ cells expressing rhodopsin per random sight. Nuclei were stained blue by DAPI. Scale bar = 200 μm.

The online version of this article includes the following figure supplement(s) for figure 6:

**Figure supplement 1.** Expression of the mutant human *RHO* allele after gene editing with SaCas9/17Sg-2 *in vitro*.

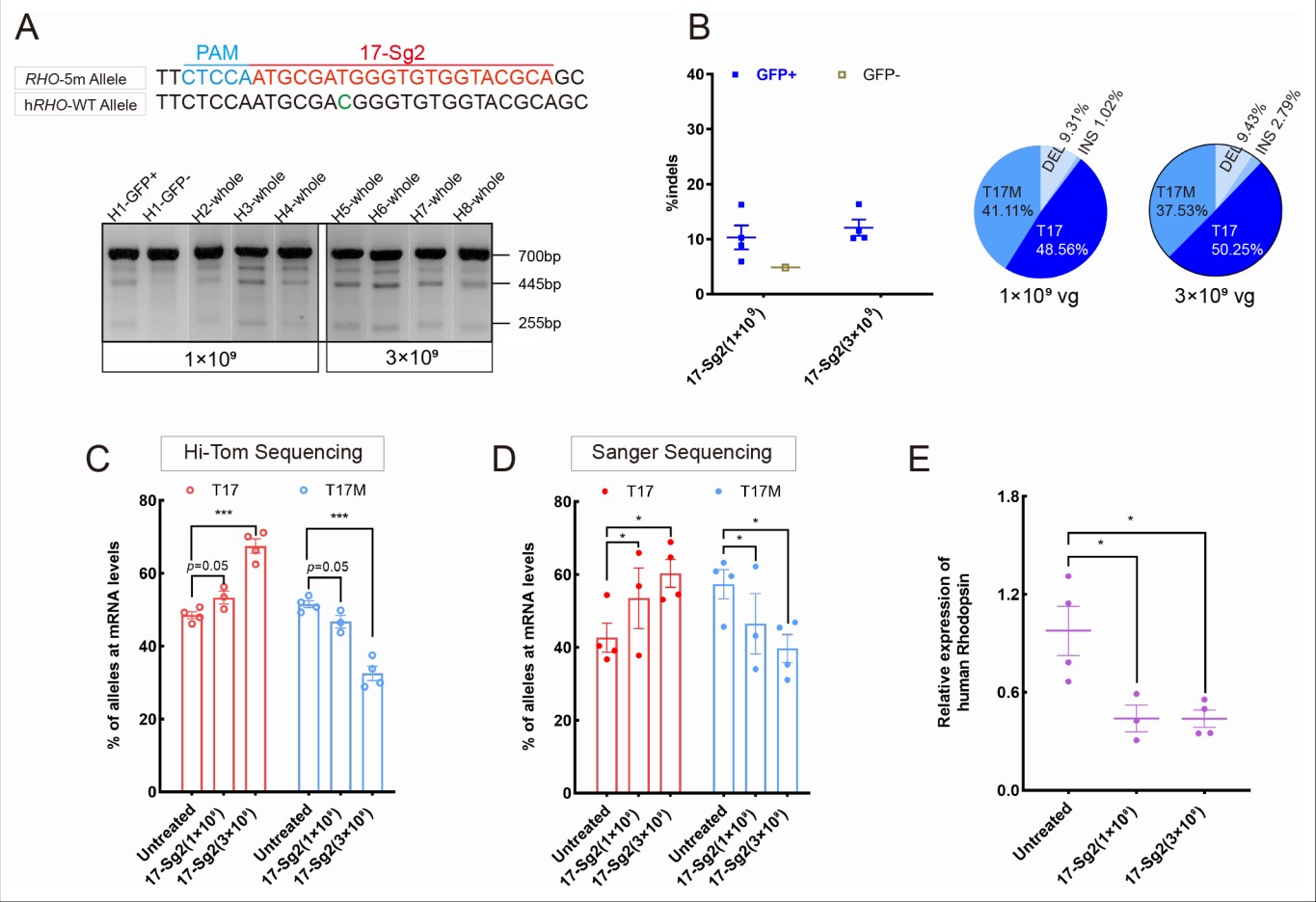

**Figure 7.** Expression of the *RHO*-5m allele after gene editing with AAV-based SaCas9/17-Sg2 in *Rho^hum/m-hum^* retinas. (**A**) T7E1 assay was performed on GFP+ and GFP- retinas to detect cutting activity of AAV-SaCas9/17-Sg2 in *Rho^hum/m-hum^* retinas at 3 mpi, the full-length amplicon was 700 bp, the two truncated amplicons were 445 bp and 255 bp, respectively; 17-Sg2 was marked in red; PAM sequence was marked in light blue and mismatches in WT human *RHO* sequence comparing to 17-Sg2 were indicated in green. (**B**) Hi-Tom sequencing was used to detect the gene-editing efficiency and types of indels in AAV-SaCas9/17-Sg2-treated *Rho^hum/m-hum^* retinas at $1×10^9$ and $3×10^9$ doses. Relative RNA level of *RHO*-WT and *RHO*-5m alleles in SaCas9/17-Sg2-treated retinas (n=3 at $1×10^9$ dose, n=4 at $3×10^9$ dose) versus age-matched untreated retinas (n=4) determined by Hi-Tom sequencing (**C**) and TA and Sanger sequencing (**D**). (**E**) Relative mRNA levels of human *RHO* in SaCas9/17-Sg2-treated retinas (n=3 at $1×10^9$ dose, n=4 at $3×10^9$ dose) versus age-matched untreated retinas (n=4) determined by qPCR analysis. Error bars show SEM, and the significance was calculated using two-tailed unpaired *t*-test, *p<0.05; ***p<0.005.

We then determined whether the indels created by SaCas9/17-Sg2 at the DNA level could result in decreased mRNA levels of the *RHO*-5m allele. AAV-EGFP and AAV-SaCas9/17-Sg2 (1:1 mixture) were co-delivered into both eyes of the *Rho^hum/m-hum^* mice, each at $1×10^9$ and $3×10^9$ doses (*Supplementary file 1b*). *Rho^hum/m-hum^* mouse, a humanized mouse model carried a human *RHO*-WT allele and *RHO*-5m allele. Truncated cleaved products treated with T7E1 nuclease were observed in treated *Rho^hum/m-hum^* retinas (*Figure 7A*), the average cutting efficiencies were 10.06% (GFP+, n=4 at $1×10^9$ dose) and 12.09% (GFP+, n=4 at $3×10^9$ dose) revealed by Hi-Tom sequencing (*Figure 7B*). Meanwhile, we employed Hi-Tom sequencing, TA and Sanger sequencing of RT-PCR products to compare the amount of mRNA transcripts of *RHO*-WT and *RHO*-5m in *Rho^hum/m-hum^* retinas at 3 mpi. At $1×10^9$ dose, Hi-Tom sequencing indicated that the average percentage of WT transcripts was higher in treated retinas than that in untreated retinas (53.31% vs 48.51%), and the average mutant transcript was lower in treated retinas than that in untreated retinas (46.69% vs 51.49%) (p=0.05; *Figure 7C*). A statistically higher percentage of WT transcripts and a lower percentage of mutant transcripts in treated retinas than the control (untreated retinas) were detected by TA and Sanger sequencing (p<0.05, *Figure 7D*). At $3×10^9$ dose, Hi-Tom sequencing, TA and Sanger sequencing indicated that the percentage of WT transcripts was statistically higher in treated retinas than that in untreated retinas (67.47% vs. 48.51%

[p<0.005] and 60.32% vs. 42.66% [p<0.05], respectively), and the mutant transcript was statistically lower in treated retinas than that in untreated retinas (32.53% vs. 51.49% [p<0.005] and 39.68% vs. 57.34% [p<0.05], respectively, *Figure 7C–D*), indicating the mRNA expression of *RHO*-5m decreased by 36.82% after treatment (Hi-Tom sequencing). The qPCR analysis indicated that the relative *RHO* mRNA expression levels were significantly lower in the treated retinas than that in untreated retinas at both $1×10^9$ (p<0.05) and $3×10^9$ doses (p<0.05, *Figure 7E*).

## Retinal functions improvement and photoreceptor preservation following the mutant *RHO* allele disruption

Scotopic electroretinogram (ERG) responses in Mut-*Rho*<sup>wt/hum</sup> mice were more affected than photopic responses (*Figure 5—figure supplement 2A*), indicating that rod dysfunction was more severe than cone dysfunction and deteriorated earlier (*Liu et al., 2022*). The ERG results were consistent with the hematoxylin and eosin (HE) and IF results, which showed that Mut-*Rho*<sup>wt/hum</sup> mice lost photoreceptors slowly, and the OS/IS length of photoreceptor cells became shorter, with increasing age (*Figure 5— figure supplement 2B–D*), indicating the pattern of retinal degeneration caused by mutant human rhodopsin was a typical rod-cone decay. Thereby, Mut-*Rho*<sup>wt/hum</sup> mice were affected with *RHO*-5m-adRP (*Liu et al., 2022*).

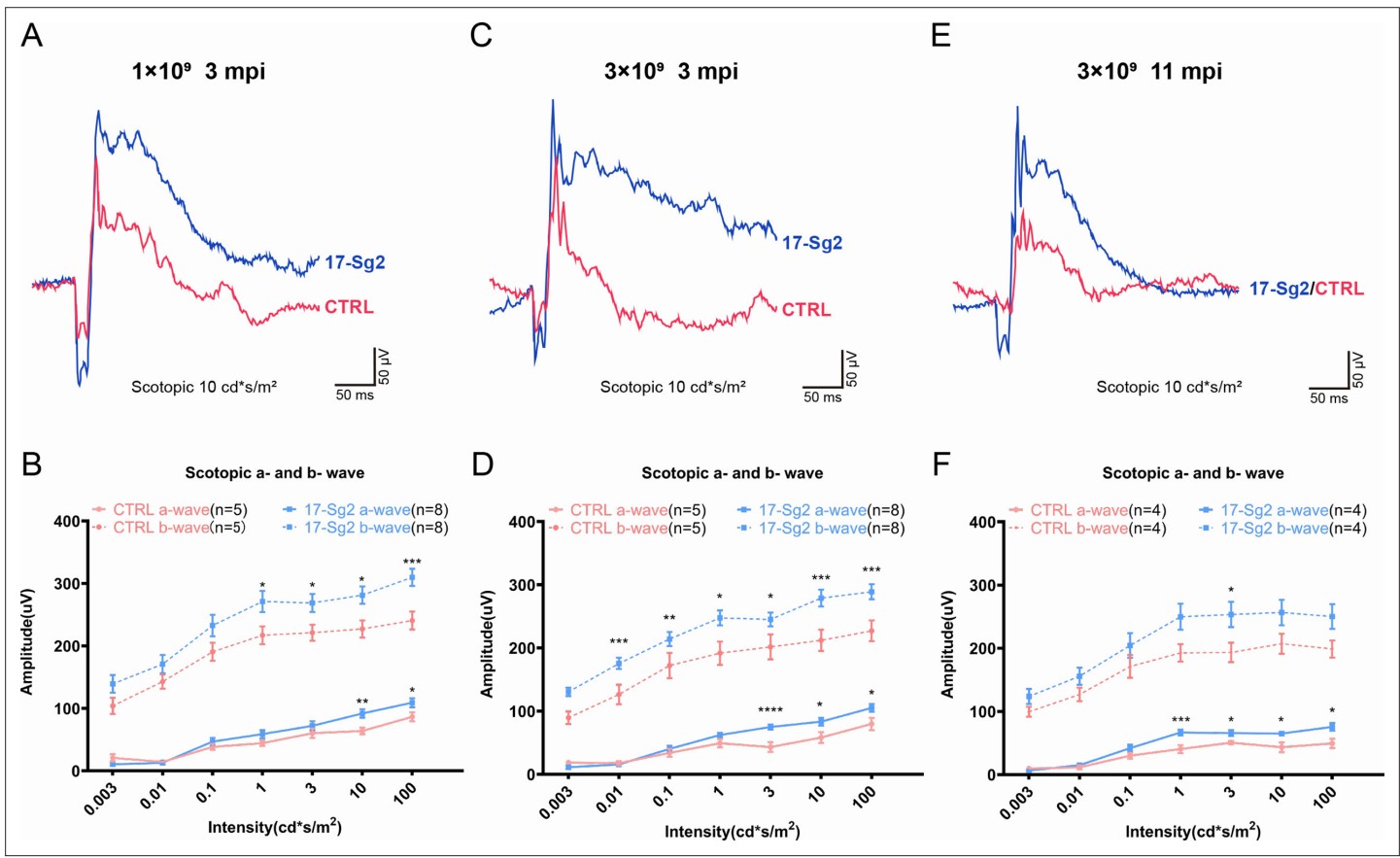

**Figure 8.** Significant improvement of retinal functions with AAV-based SaCas9/17-Sg2 in Mut-*Rho*<sup>wt/hum</sup> mice determined by full-field ERG. Representative scotopic ERG (at the flash intensity of 10 cd*s/m²) waveforms of Mut-*Rho*<sup>wt/hum</sup> mice treated with AAV-SaCas9/17-Sg2 (blue) and AAV-SaCas9/CTRL (red) at $1×10^9$ (3 mpi, **A**), $3×10^9$ dose (3 mpi, **C**), and at $3×10^9$ dose (11 mpi, **E**). Light dependence profile of scotopic a- and b-wave amplitudes of Mut-*Rho*<sup>wt/hum</sup> mice treated with AAV-SaCas9/17-Sg2 (blue) and AAV-SaCas9/CTRL (red) at $1×10^9$ dose (**B**, 3 mpi), $3×10^9$ dose (**D**, 3 mpi) and at $3×10^9$ dose (**F**, 11 mpi). Error bars show SEM, and the significance was calculated using two-tailed unpaired *t*-test, *p<0.05; **p<0.01; ***p<0.005.

The online version of this article includes the following figure supplement(s) for figure 8:

**Figure supplement 1.** The ERG results of AAV-SaCas9/17-Sg12-treated Mut-*Rho*<sup>wt/hum</sup> mice (n=8), AAV-SaCas9/CTRL-treated mice (n=5), untreated Mut-*Rho*<sup>wt/hum</sup> mice (n=5), and untreated WT mice (n=7).

To assess whether the retinal functions in treated mice improved, AAV-EGFP and AAV-SaCas9/17-Sg2 (1:1 mixture) or AAV-EGFP and AAV-SaCas9/CTRL (1:1 mixture) were co-delivered into both eyes of Mut-*Rho*^wt/hum mice, each at $1\times10^9$ and $3\times10^9$ dose (*Supplementary file 1b*). At 3 mpi, ERG analysis showed a significant improvement in scotopic a- and b-waves in retinas treated with AAV-SaCas9/17-Sg2 compared to their corresponding controls at different stimulus intensities at doses of $1\times10^9$ or $3\times10^9$ (*Figure 8A–D*, *Figure 8—figure supplement 1*). Similarly, at 11 mpi, at $3\times10^9$ dose, ERG analysis also indicated a significant improvement in scotopic a- and b-waves in retinas treated with AAV-SaCas9/17-Sg2 compared to their corresponding controls at several stimulus intensities (*Figure 8E–F*). Besides, the retinal functions of AAV-SaCas9/17-Sg2-treated Mut-*Rho*^wt/hum mice were better than that of untreated age-matched Mut-*Rho*^wt/hum mice, and had no significant difference when compared to that of untreated same-age WT mice (*Figure 8—figure supplement 1*).

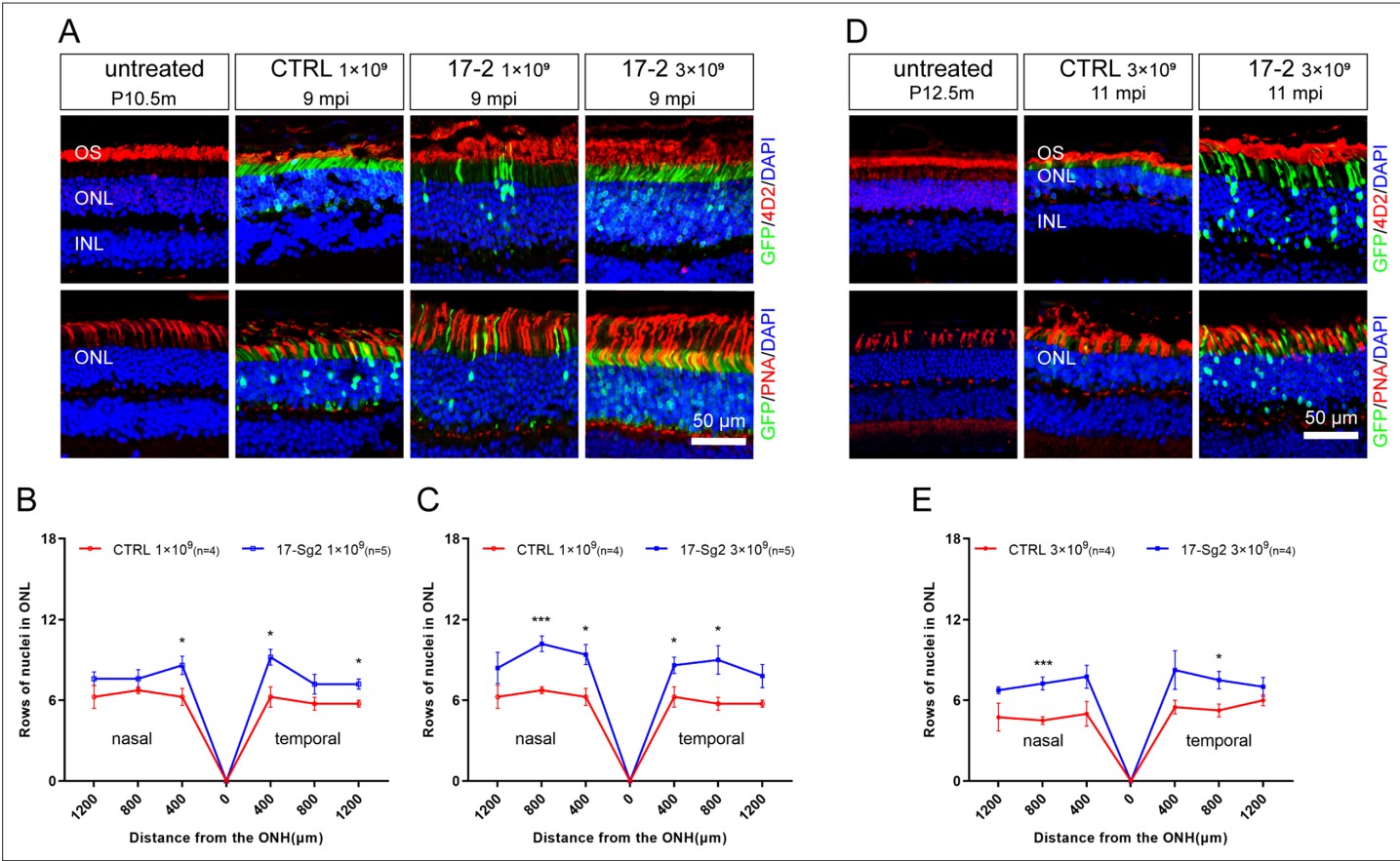

**Figure 9.** Photoreceptor cell preservation after gene editing with AAV-based SaCas9/17-Sg2 in Mut-*Rho*^wt/hum mice. (**A**) At 9 mpi, IF staining of rhodopsin (4D2, red) and PNA (red) from age-matched untreated mouse retina, the AAV-SaCas9/CTRL-treated and AAV-SaCas9/17-Sg2-treated retinal regions (GFP+) in Mut-*Rho*^wt/hum mice at $1\times10^9$ and $3\times10^9$. The graph revealed the average number of rows of ONL nuclei in AAV-SaCas9/17-Sg2-treated (blue) and AAV-SaCas9/CTRL-treated (red) Mut-*Rho*^wt/hum mice at $1\times10^9$ (**B**) and $3\times10^9$ (**C**) dose at 9 mpi. (**D**) At 11 mpi, IF staining of rhodopsin (4D2, red) and PNA (red) from age-matched untreated mouse retina, the AAV-SaCas9/CTRL-treated and AAV-SaCas9/17-Sg2-treated retinal regions (GFP+) in Mut-*Rho*^wt/hum mice at $3\times10^9$. (**E**) The graph revealed the average number of rows of ONL nuclei in AAV-SaCas9/17-Sg2-treated (blue) and AAV-SaCas9/CTRL-treated (red) Mut-*Rho*^wt/hum mice at $3\times10^9$ dose at 11 mpi. Error bars show SEM, and the significance was calculated using two-tailed unpaired *t*-test, *p<0.05; ***p<0.005. Rhodopsin and PNA were indicated in red, nuclei were stained blue by DAPI. ONL = outer nuclear layer; OS = outer segment; IS = inner segment. P10.5/12.5m = postnatal 10.5/12.5 month. Scale bar = 50 μm.

The online version of this article includes the following figure supplement(s) for figure 9:

**Figure supplement 1.** Retinal images after gene editing with AAV-based SaCas9/17-Sg2 in Mut-*Rho*^wt/hum mice at 4 mpi.

**Figure supplement 2.** Flat-mount images of Mut-*Rho*^wt/hum retinas after treatment.

**Figure supplement 3.** Retinal images after gene editing with AAV-based SaCas9/17-Sg2 in Mut-*Rho*^wt/hum mice.

**Figure supplement 4.** Inner retinal cells preservation after gene editing with AAV-based SaCas9/17-Sg2 in Mut-*Rho*^wt/hum mice.

To evaluate whether silence or ablation of the *RHO*-5m allele could result in prevention or delay of photoreceptor cells loss, we evaluate the retinal structure in AAV-SaCas9/17-Sg2- and AAV-SaCas7/CTRL-treated Mut-*Rho*$^{wt/hum}$ mice at 4 mpi, 9 mpi, and 11 mpi, at $1\times10^9$ and/or $3\times10^9$ dose (***Supplementary file 1b***). The average number of rows of ONL nuclei of AAV-SaCas9/17-Sg2-treated retinas was not significantly different from that of AAV-SaCas7/CTRL-treated retinas at 4 mpi (***Figure 9—figure supplement 1***), either at $1\times10^9$ or $3\times10^9$ doses. Later at 9 mpi, either at $1\times10^9$ or $3\times10^9$ doses, measurement of the average number of rows of ONL nuclei revealed that there were significantly more photoreceptor cells in AAV-SaCas9/17-Sg2-treated retinas than that in age-matched untreated retinas (***Figure 9A–C***, ***Figure 9—figure supplement 2A–C***), IF staining showed that SaCas9/17-Sg2-treated Mut-*Rho*$^{wt/hum}$ mice had longer OS/IS (***Figure 9A***), which retained until at least 11 mpi (***Figure 9D–E***, ***Figure 9—figure supplement 2D–E***). Our previous study showed that the mutant rhodopsin localized to RIS and cell bodies (***Liu et al., 2022***). As shown in ***Figure 9—figure supplement 1A and C***, compared to CTRL groups, mutant rhodopsin in RIS and rods bodies (4D2, red) decreased significantly in AAV-SaCas9/17-Sg2-treated mouse retinas, indicating the mutant rhodopsin was ablated in rods, similar results were also detected at 9 mpi and 11 mpi (***Figure 9—figure supplement 3***). Besides, examples of mouse cryosections labeled with various cell-specific markers for IF microscopy were also given in ***Figure 9—figure supplement 4***, protein kinase C (PKC-α) for rod bipolar cells, calbindin for horizontal cells, and CRALBP for RPE and Müller cells which span throughout the retina vertically, compared to untreated mice and SaCas9/CTRL-treated mice at 9 mpi and 11 mpi, more and healthier bipolar cells, horizontal cells and Müller cells were detected in AAV-SaCas9/17-Sg2-treated retinas. These results showed clearly that AAV-SaCas9/17-Sg2 effectively restored the retinal function and preserved photoreceptor cells in the humanized *RHO* mutant mouse model, supporting the potential of AAV-SaCas9/17-Sg2 as an effective drug for *RHO*-T17M-associated RP.

## Analysis of off-target activity and vector safety

To evaluate the off-target activity of SaCas9/17-Sg2 in human gDNA, possible off-target sites obtained from the Benchling's CRISPR Tool (***Supplementary file 1d***) were analyzed. The T7E1 assay showed no detectable off-target activity for these candidate sites for SaCas9/17-Sg2 (OT1-OT10, ***Figure 10—figure supplement 1***), which was consistent with the results of Sanger sequencing (data not shown). Besides, the top 20 off-target sites of 17-Sg2 were also predicted by the Cas-OFFinder online tool (***Supplementary file 1e***), comparing the NGS results of PCR amplicons amplified from 293T cells which were transfected with 17-Sg2 plasmid and CTRL plasmid, we observed no detectable off-target activity for these candidate sites (***Supplementary file 1f***). As illustrated in ***Supplementary file 1f***, considering that SaCas9/17-Sg2 targeted *RHO*-T17M allele specifically, the corresponding *RHO*-WT sequence from 293T was supposed to be a specific off-target site, thereby the genomic region that was flanking the specific off-target site was also amplified and sequenced, comparing the NGS results of 293T cells transfected with 17-Sg2 plasmid and CTRL plasmid, no detectable off-target activity for this site was detected. Together, these results confirmed the allele-specific gene-editing activity of SaCas9/17-Sg2. To further examine the off-target potential in human gDNA resulting from SaCas9/17-Sg2, we performed whole-genome sequencing (WGS) in 293T cells transfected with 17-Sg2 plasmid and the control 293T cells (without transfection). We analyzed the single-nucleotide variants (SNVs) and indels in two cell groups. A total of 3,336,802 SNVs and 1,361,010 indels were detected in 17-Sg2 cells, which were not much different from the control (3,367,623 SNVs and 1,383,287 indels) (***Figure 10***). We then filtered the SNVs and indels using Cas-OFFinder-predicted off-target sites and the WGS result for 17-Sg2 cells and untreated cells. We found 1 SNVs and 8 indels (***Supplementary file 1g and h***), but none of them were located in coding regions, suggesting no functional off-target sites in SaCas9/17-Sg2-treated human gDNA.

To further analyze the safety of this gene-editing drug, four adult, female nonhuman primate (NHP, *Macaca fascicularis*) were employed. One of them was untreated, while AAV-SaCas9/17-Sg2 were also delivered to bilateral subretinal space of three NHPs ($3\times10^{11}$ vg/ eye, 100 µL). At 92 days post injection (92 dpi), qPCR analysis revealed that 17-Sg2 and SaCas9 were expressed in NHP retinas (***Figure 10—figure supplement 2A***). Scanning laser ophthalmoscope-evoked multifocal ERG (SLO-mfERG) analysis indicated that the gene-editing drug would not damage NHP's retinal functions (***Figure 10—figure supplement 2B***). Optical coherence tomography (OCT) analysis and HE staining

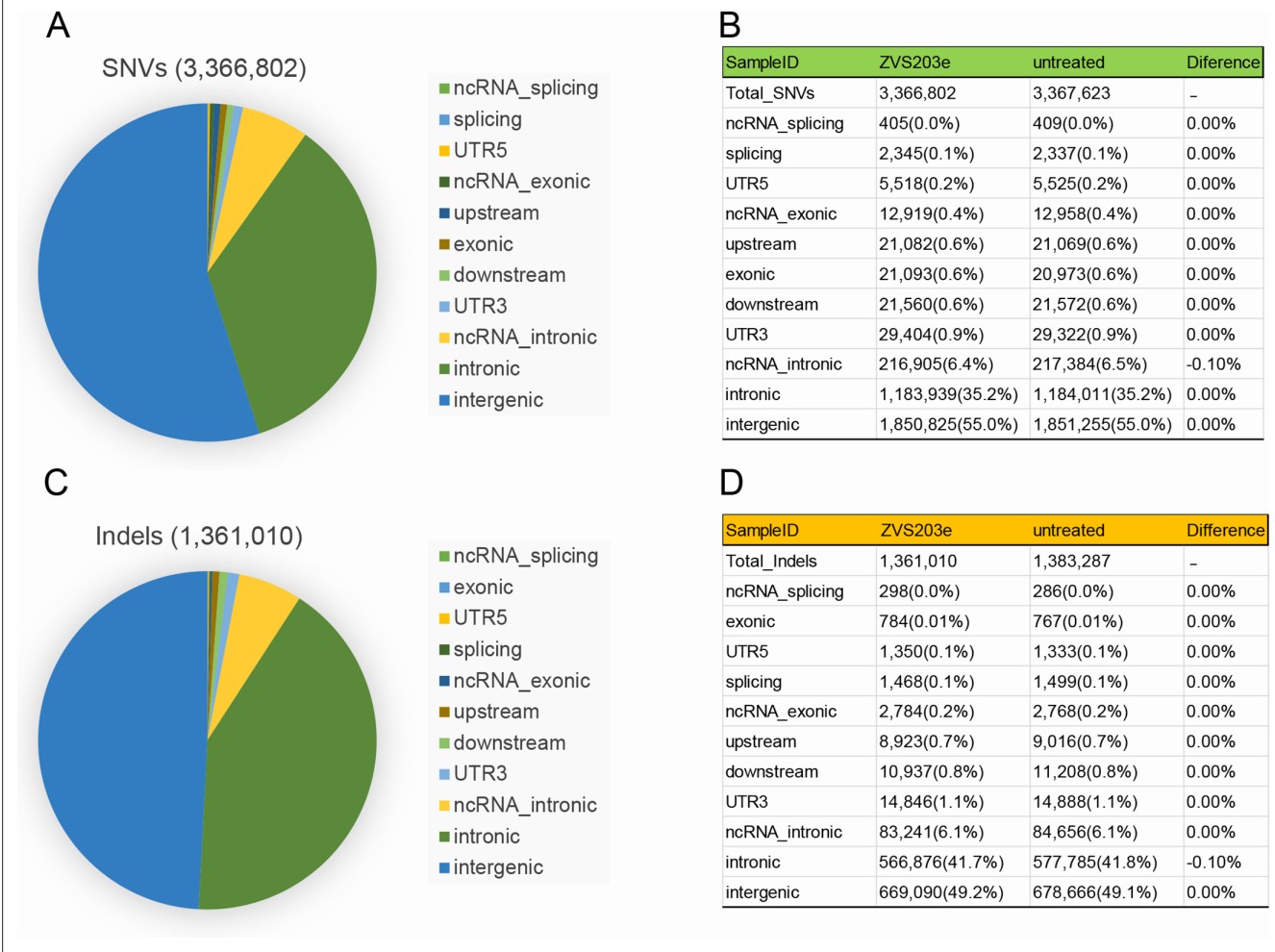

**Figure 10.** Examination of SaCas9/17-Sg2 off-target effects in human gDNA using WGS. Identification of SNVs (**A**) and indels (**C**) in 293T cells transfected with 17-Sg2 plasmid at the WGS level. The type of SNVs (**B**) and indels (**D**) in 293T cells transfected with 17-Sg2 plasmid and untreated cells at the WGS level.

The online version of this article includes the following figure supplement(s) for figure 10:

**Figure supplement 1.** Analysis of SaCas9/17-Sg2 off-target activity.

**Figure supplement 2.** Safety analysis of AAV-SaCas9/17-Sg2 in NHP.

indicated that the gene-editing drug would not damage NHP's retinal structures (*Figure 10—figure supplement 2C–D*).

## Discussion

There is currently an increasing number of gene-editing clinical trials to treat hereditary diseases such as sickle cell disease, β-thalassemia (*Frangoul et al., 2021*) and transthyretin amyloidosis (*Gillmore et al., 2021*), and even cancer (*Stadtmauer et al., 2020*). These published data demonstrated the safety and feasibility of the CRISPR/Cas9 system *in vivo*. Previously, scientists have contributed significantly to the treatment of *RHO*-adRP using the allele-specific CRISPR/Cas9 gene-editing strategy in iPSCs and animal models (*Behnen et al., 2018*; *Bakondi et al., 2016*), demonstrating the promising prospect of this approach in developing effective treatment for this disease. However, none of these methods can be directly applied in clinical practice. Herein, we developed an allele-specific gene-editing therapeutic for *RHO* c.50C>T (p.T17M) for the first time. SaCas9/17-Sg2 was highly active and specific to the human *RHO* target sequence. *In vitro* experiments using 293T cells and patient-specific iPSCs have demonstrated strong nuclease specificity. A single AAV2/8 containing the SaCas9 and

17-Sg2 could be directly used as a therapeutic drug for adRP caused by the *RHO*-T17M mutation. This potential application was experimentally tested by subretinal delivery of AAV2/8-EFS-SaCas9-U6-17-Sg2 into the *RHO* humanized mice. Analysis of the treated retinas clearly showed its long-term therapeutic effect (up to 11 mpi) including retinal function improvement and photoreceptor preservation in the heterozygous mutant humanized mice (*Liu et al., 2022*). Moreover, there was no unwanted off-target activities observed. Thus, our study makes a significant step toward the development of an effective drug for *RHO*-adRP.

Compared with allele-specific gene-editing therapy, allele-independent gene ablation and replacement approaches are more cost-effective and efficient. However, there are potential risks with the gene ablation and replacement strategy. First, prior studies show that there is a fine balance between insufficiency and toxicity of rhodopsin (*Lewin et al., 2014*). Changes in the rhodopsin level (either suppression of rhodopsin expression or overexpression) may alter the morphology of rod OS (ROS) and discs (*Makino et al., 2012*; *Wen et al., 2009*). It is difficult for the ablation and replacement approach to strike such a balance *in vivo*. Second, previous studies have indicated that $Rho^{-/-}$ mice lose all of their rod photoreceptors extremely rapidly (approximately 3 months) (*Humphries et al., 1997*; *Hobson et al., 2000*), implying that disruption of both WT and mutant *RHO* alleles may aggravate the disease, when the exogenous augmentation of WT cannot rescue this disruption. Allele-specific silencing of mutant rhodopsin at the RNA or DNA levels can avoid the above problems. Although rhodopsin function is thought to be essential for the formation of ROS and phototransduction initiation (*Gunkel et al., 2015*), heterozygous rhodopsin-knockout mice showed little retinal degeneration, indicating that one copy of this gene is sufficient for normal retinal development and functions (*Lem et al., 1999*). In this respect, maximal therapeutic benefit is expected when the mutant allele is completely ablated, while the normal allele is functionally intact. However, many factors may affect therapeutic effects. For instance, ASO can deregulate mutant rhodopsin levels transcriptionally, but the degree to which the toxic protein must be reduced to eliminate any disease-causing effect is not known (*Jun Wan, 2016*). The presence of gain-of-function mutations and repeated subretinal injection may diminish the effectiveness of treatment. The CRISPR/Cas9 gene-editing strategy in mammalian cells can provide a means to delete or correct disease-causing mutations while leaving the gene under the control of its endogenous regulatory elements (*Jinek et al., 2013*; *Mali et al., 2013*; *Burnight et al., 2018*), making this approach also promising for the treatment of inherited diseases.

CRISPR/Cas9 gene-editing therapy is currently being advanced as the treatment of human IRDs (*Maeder et al., 2019*; *Burnight et al., 2018*). A study conducted by *Maeder et al., 2019*, shows that one of the complexities of drug development for gene editing is that it is human genome-specific. Herein, we used human cells and the *RHO* humanized mice to demonstrate that SaCas9/17-Sg2 could reach efficient and allele-specific gene editing, thereby improving the retinal functions and delaying photoreceptor cells loss. We also proved that SaCas9/17-Sg2 did not result in off-target activity using WGS. *In vitro* patient-derived iPSCs generated in this study offered as an ideal research material to validate gene-editing specificity and cutting efficiency at the DNA level. Moreover, 3D retinal organoids or rod cell precursors derived from iPSCs (*Capowski et al., 2019*) have the potential to be used to mimic disease phenotypes or for autologous transplantation (*Assawachananont et al., 2014*; *Nozie, 2017*; *Santos-Ferreira et al., 2016*) after gene editing. However, we must emphasize that the transfection and gene-editing efficiency in iPSCs are evidently lower than those of some commonly used cell lines (*Burnight et al., 2017*; *Lin et al., 2014*), which may be due to the significant cell death that probably results from an overwhelming cytosolic dsDNA-induced innate immune response (*Sun et al., 2013*; *Kim et al., 2018*), implying that the on-target and off-target effects in iPSCs might be underestimated.

The first factor that may raise safety concerns for allele-specific CRISPR/Cas9 drug development is the specificity of SaCas9/SgRNA (*Maeder et al., 2019*). This study demonstrated that SaCas9/17-Sg2 did not disrupt human WT *RHO* both *in vitro* and *in vivo*. In fact, a large number of allele-specific sgRNAs can be designed for disease-causing mutations in *RHO* (over 200 mutations, HGMD), such as the three sgRNAs for T17M. However, not all of them work well. A previous study proves theoretically and practically that the specificity of sgRNA depends on its seed sequence, which is a stretch of 8–14 nucleotide (nt) immediately upstream of the PAM. Even point mutations within the seed sequence can abolish cleavage by SpCas9 nucleases (*Hsu et al., 2013*; *Jiang et al., 2013*; *Semenova et al., 2011*), and spCas9 nuclease tolerates single-base mismatches in the PAM-distal region to a greater

extent than in the PAM proximal region (*Hsu et al., 2013*; *Jinek et al., 2012*). Herein, the specificity of 17-Sg1 and -Sg2 are better than that of 17-Sg3, may due to the fact that the point mismatch (c.50C>T) is within the seed sequence of 17-Sg1 and -Sg2 but not within that of 17-Sg3 (*Figure 1*), showing that the targeting specificity of SaCas9/SgRNA is also dependent on the seed sequence. Previous studies have indicated that T17M rhodopsin proteins fail to fold correctly, are retained in the ER, and cannot easily reconstitute with 11-cis-retinal (*Mendes et al., 2005*; *Behnen et al., 2018*). Our experiments in Hela cells and 293T cells indicates that rhodopsin protein with T17M and five variants are retained in cytoplasm and the endoplasmic reticulum (*Liu et al., 2022*). Another factor which may raise safety concerns is that indels created by gene editing possibly result in new rhodopsin variants which may also exert toxic effect or dominant-negative on WT rhodopsin (*Patrizi et al., 2021*). *In silico* analysis and detailed *in vitro* experiments indicate that *RHO*-5m mRNA with these most frequent indels generated by gene editing near the N-terminal of *RHO* gene can be degraded due to the effect of NMD (*Figure 6* and *Figure 6—figure supplement 1*). Work conducted by *Luo et al., 2021*, suggested that most of the truncated variants involving upstream of K296 are likely benign, as shown in *Supplementary file 1c*, all frameshift indels created by AAV-SaCas9/17-Sg2 are upstream of K296, thereby the truncated products after gene editing are benign. Experiments in 293T cells show that this insertion can lead to NMD of the *RHO*-5m mRNA, consisting with the above theory. These results support the potential safety of gene editing of the N-terminal domain of rhodopsin. Additionally, the preclinical results in NHP further confirmed the safety of the gene-editing drug.

Currently, there are few available animal models compared to the number of known *RHO* mutations (over 200). Animals with human gene knock-in allow us to study and model disease, and permit testing of the therapeutic effect of CRISPR/Cas9 in preclinical *in vivo* systems, without putting humans at risk (*Hosur et al., 2017*). The shortage of humanized animal models may hamper the application of CRISPR/Cas9 technologies from animals toward patients. In the past, several *RHO* humanized mouse models were bred to express mutant human rhodopsin in a heterozygous knockout mouse rhodopsin (*Rho*$^{+/-}$) or WT background (*Lem et al., 1999*; *Chan et al., 2004*; *Olsson et al., 1992*) by introducing human *RHO* genomic fragments that encompassed the entire transcriptional unit as well as its upstream and downstream flanking sequences. In contrast, our humanized models are generated by replacing the mouse *Rho* gene with the human *RHO* gene, but not interrupting the flaking sequencing of this gene. Previous studies (*Mendes et al., 2005*; *Liu et al., 2021*) and data from HGMD have reported that each of the five heterozygous mutations in *RHO* can cause adRP. Our study indicates (*Liu et al., 2022*) that the typical adRP phenotypes of Mut-*Rho*$^{wt/hum}$ mice result from the *RHO*-5m allele, and the slow disease progression offers us the opportunity to observe therapeutic effects after treatment. Importantly, mice with different genotypes can be obtained from the intercross of these models according to our needs, thereby facilitating our *in vivo* experiments. The humanized mouse models combined with our strategy to selectively target the mutant *RHO* allele faithfully demonstrate their application prospects as well.

Recently, AAV has been considered an ideal choice for therapeutic gene delivery in the retina and has been confirmed to be promising for the delivery of CRISPR/Cas9 reagents (*El Refaey et al., 2017*; *Yang et al., 2016*). Besides Luxturna, many gene therapy clinical trials based on AAVs are ongoing (https://www.clinicaltrials.gov/), indicating the feasibility and safety of AAVs. However, AAVs have a relatively small packaging capacity (~4.7 kb), which limits the size and number of sequences that can be incorporated into a single virus (*Burnight et al., 2018*; *Ran et al., 2015*). To put all components into one AAV vector, the cytomegalovirus (CMV) enhancer promoter is replaced by the EFS promoter, and SaCas9, which is much smaller than SpCas9 (*Ran et al., 2015*), is chosen and packed in an AAV vector with its sgRNA. As long as 9 mpi, no detectable toxic effects in mouse retinas are identified at three different doses when comparing the retinal structures of treated and untreated mice. Gene-editing efficiencies herein are dose-dependent, yet the specificity of AAV-SaCas9/17-Sg2 is not affected by increasing AAV dose. However, indel analysis by Sanger sequencing in mouse retinas after treatment showed that a 49 bp insertion from the AAV inverted terminal repeat (ITR) sequence occurs at the cut site at a dose of $3 \times 10^9$. Previous studies confirm that the majority of AAV genome insertions consist of an ITR sequence (*Jarrett et al., 2017*), which might be due to the combined properties of AAV ITR and its interaction with NHEJ host-cell machinery (*McCullough et al., 2019*; *Young and Samulski, 2001*). In this study, no AAV ITR insertions in treated retinas were detected in the $5 \times 10^8$ and $1 \times 10^9$ dose groups. We consider whether AAV dose will influence the possibility of

insertions in question, and more research should be conducted to validate this hypothesis. At either DNA or RNA levels, the therapeutic effect in the retina at the $3×10^9$ dose was better than that at the $1×10^9$ dose. As shown in *Figure 8—figure supplement 1*, at 3 mpi, the retinal functions of untreated WT and Mut-*Rho^{wt/hum}* mice were significantly better than that of SaCas9/17-Sg2- and SaCas9/CTRL-treated mice, while the remarkable long-term therapeutic effect in SaCas9/17-Sg2-treated mice was detected, implying that: (a) compared to NHP and human eyes, the eye volume of mouse is relatively small, subretinal injection which is an invasive operation will result in retinal detachment, certainly leading to a damage to retinal functions. However, as times go on, retinal detachment will recover, exerting less effect on the retinal functions, indicating that the long-term therapeutic effect of drug should be observed. (b) In preclinical experiments or even clinical trials, one variable between treated group and control group (gene-editing drug with 17-Sg2 or not) is essential, to ensure the statistical results reliable. For instance, between untreated Mut-*Rho^{wt/hum}* mice and treated ones, there were two variables: (a) subretinal injection, which is an invasive operation, (b) the gene-editing drug including AAV-SaCas9/17-Sg2 or not. Previous studies (*Mahfouz et al., 2011*; *Hsu et al., 2013*) have indicated that enzymatic specificity and activity strength are often highly dependent on reaction conditions, high enzyme concentration will amplify off-target activity, and one potential strategy for minimizing nonspecific cleavage is to limit the enzyme concentration. Therefore, the balance between therapeutic efficiency and AAV dose of subretinal injection *in vivo* must be weighted. Since approximately 10% of functional foveal cone photoreceptors are sufficient for near-normal visual acuity, researchers hypothesize that correction of the *CEP290* IVS26 mutation in at least 10% of foveal cones will lead to clinical benefit in patients (*Maeder et al., 2019*). The average cutting efficiencies in GFP+ retinas at $1×10^9$ or $3×10^9$ doses are more than 10%, and the long-term therapeutic effect in more treated mice should be evaluated to obtain more unbiased results.

In summary, this study demonstrates the safety and effectiveness of our allele-specific gene-editing therapy *in vitro* and in humanized mouse models, thereby offering a generalizable framework for the preclinical development of gene-editing medicine and future clinical trials.

## Materials and methods

### Animals

Approval was obtained from the Peking University Health Science Center Ethics Committee for Experimental Animal Research (Research License LA20200473). All procedures were performed according to the regulations of the Association for Research in Vision and Ophthalmology's statement for the use of animals in ophthalmic and vision research. All mice were maintained in accordance with the guidelines of the Association for the Assessment and Accreditation of Laboratory Animal Care.

*RHO* humanized mouse models (with/without *RHO*-5m allele) with a C57BL/6J background using HDR and CRISPR/Cas9 strategy were custom designed and obtained from Beijing Biocytogen Co., Ltd (Beijing, China). Mice were bred and maintained at the Peking University Health Science Center Animal Care Services Facility under specific pathogen-free conditions with a 12 hr light/12 hr dark cycle. Food and water were provided ad libitum.

Four adult, female NHP were bred and maintained at JOINN Laboratories (Suzhou, China), approval was obtained from the JOINN Laboratories Ethics Committee for Experimental Animal Research (Research License ACU21-1108).

### Patients

The study was approved by the Medical Scientific Research Ethics Committee of Peking University Third Hospital (Research License 2021262). The procedures were performed in accordance with the tenets set forth in the Declaration of Helsinki. All patients provided written informed consent for this study.

### Generation of patient-specific iPSCs

Urine samples from a male patient (45 years of age) affected with adRP resulting from *RHO*-T17M mutation (*Liu et al., 2021*) and a healthy male control (27 years of age) were collected. UCs were isolated using a UC isolation medium (CA3102500, Cellapy) and cultured for two passages in a UC expansion medium (CA3103200, Cellapy). UCs were reprogrammed using the integration-free CytoTune

2.0 Sendai Reprogramming Kit (A16517, Thermo Fisher Scientific, MA, USA), which contained four Yamanaka factors, SOX2, OCT3/4, KLF4, and c-MYC, according to the manufacturer's instructions. After the emergence of colonies with iPSC-like morphology, R1 medium (85851, STEMCELL) was used for the culture at 37°C under 5% $CO_2$ conditions. The primary antibodies and corresponding secondary antibodies for IF staining were listed below (*Zhou et al., 2012*; *Takahashi et al., 2007*; *International Stem Cell Initiative et al., 2007*).

| Primary antibodies | Corresponding secondary antibodies |
| --- | --- |
| Rabbit polyclonal anti-Oct4 (1 µg/mL, ab19857, Abcam) | Alexa Fluor 568 donkey anti-rabbit IgG (1:800 dilution, A10042, Abcam) |
| mouse monoclonal anti-TRA-1–60 (1:500 dilution, ab16288, Abcam) | Alexa Fluor 488 donkey anti-mouse IgG (1:800 dilution, A21201, Abcam) |
| Rabbit polyclonal anti-Nanog (1:1000 dilution, ab21624, Abcam) | Alexa Fluor 568 donkey anti-rabbit IgG (1:800 dilution, A10042, Abcam) |
| Mouse monoclonal anti-SSEA-4 (1:500 dilution, sc-21704, Santa Cruz Biotech) | Alexa Fluor 488 donkey anti-mouse IgG (1:800 dilution, A21201, Abcam) |

## SgRNA design and *in vitro* cleavage efficiency study

Oligo pairs of 21 nt sgRNA sequences targeting the *RHO*-T17M mutation were designed based on the Benchling website (https://www.benchling.com/). Based on the *RHO* gene sequence (ENSG00000198947), we identified all the NNGRR PAMs for SaCas9. Oligonucleotides corresponding to the sgRNA of interest were purchased from Beijing Ruibio Biotech Co., Ltd. (Beijing, China).

An *in vitro* cutting assay using the SaCas9/SgRNA Cutting Efficiency Detection Kit (VK012, Viewsolid Biotech, Beijing, China) was performed to ensure the relative cutting efficiency and specificity of sgRNAs. The sgRNAs of interest were directly transcribed in vitro with T7 polymerase from double-stranded DNA templates. Primer pairs RHO17-F and R (*Supplementary file 1i*) were used to amplify the targeted region with or without c.50C>T mutation, yielding a 678 bp PCR fragment, respectively. Afterward, 1 µL of purified PCR amplicon (50 ng/µL), 1 µL of in vitro-transcribed sgRNA (50 ng/µL), 1 µL of SaCas9 protein (1 U/µL), 2 µL of 10× SaCas9 buffer, and 15 µL of ddH₂O were added into a tube and incubated for 30 min at 37°C. Subsequently, 2 µL of DNA loading buffer (9157, TAKARA, Kusatsu, Japan) was added to the reaction and incubated at 65°C for 5 min. The full reaction was run on 2% agarose gel (111860, Biowest).

## Vector constructs

The selected sgRNA was cloned into the plasmid pX601-AAVCMV: NLS-SaCas9-NLS-3xHA-bGHpA; U6: BsaI-sgRNA (pX601, #61591, Addgene). The CMV enhancer promoter was replaced by a commonly used short, ubiquitous promoter-EFS. The plasmid pX601-EFS-SaCas9 was linearized using BsaI restriction enzyme (R3733S, New England Biolabs, MA, USA), followed by purification with a gel extraction kit (D2500, Omega). The sgRNA of interest was cloned into the BsaI site using T4 DNA ligase (M0202S, NEB), and then transformed into Top10-competent bacteria. Clones were amplified in a liquid Luria-Bertani broth, followed by the extraction of plasmid DNA (D6943, OMEGA). In addition, the plasmid containing the puromycin selection cassette AAV-EFS-SaCas9-p2a-Puro was generated from pX601-EFS-SaCas9, and the selected sgRNA was cloned into this plasmid or into Lenti_SaCRISPR_GFP (ATCC, #118636). In replacing the SaCas9 sequence with the EGFP sequence, the pX601-EFS-GFP plasmid was generated.

For the rhodopsin overexpression experiments, a full-length *RHO* cDNA with or without the T17M mutation was cloned into a lentiviral expression vector FUGW (Addgene, #14883), and the *RHO* CDS was inserted under the control of the UbC promoter. The full-length *RHO* cDNA with or without the T17M mutation was also cloned into the pmCherryN1 and pEGFPN1 vectors using the pEASY-Basic Seamless Cloning and Assembly Kit (CU-201, TransGen Biotech, Beijing, China) according to the manufacturer's protocol. *RHO* variant plasmids (*RHO*-c.46delG/c.45_46delTG/c.43_46del4bp/c.46_47ins49bp) based on the *RHO*-T17M-pEGFPN1 or *RHO*-5m-pEGFPN1 were generated. The primer pairs used were listed in *Supplementary file 1i*.

## Viral production

For the production of lentivirus, 293T cells were transfected with a combination of three plasmids, FUGW-*RHO*-cDNA or Lenti_SaCRISPR_GFP, Pax2, and vesicular stomatitis virus G protein (VSV-G) plasmid using polyetherimide (PEI) (B600070, ProteinTech Group, Chicago, IL, USA) according to the manufacturer's protocol. Twenty-four hours after transfection, the medium was replaced with a fresh one. Forty-eight hours later, the medium containing the viral particles was collected.

The AAV2/8 vectors that encoded the SaCas9 gene under the control of the EFS promoter and the sgRNA of interest under the control of the U6 promoter (AAV2/8-EFS-SaCas9-U6-SgRNA, AAV-SaCas9/SgRNA) were produced in 293T cells by PEI co-transfection with three different plasmids: pX601-EFS-SaCas9-U6-sgRNA plasmid (SgRNA plasmid), packaging plasmid pAAV-RC8 for the *rep* and *cap* genes, and the pHelper plasmid (Cell Boil, Inc). Furthermore, the AAV-EGFP from the pX601-EGFP plasmid and the AAV-SaCas9/CTRL plasmid from the CTRL plasmid were also produced. The titers were determined by qPCR.

## Cell cultures and plasmids transfection or viral infection

293T cells (CRL-3216, ATCC) were purchased from ATCC and cultured in Dulbecco's modified Eagle's medium (CM15019, Macgene, Beijing, China) and 10% fetal bovine serum (10099-141C, Gibco, USA). The cells were authenticated using STR profiling and tested negative for mycoplasma contamination. For viral transduction, the medium was removed and replaced with a fresh one as well as the previously purified lentivirus. On the day after transduction, the cells were washed with 1× DPBS (CC010, Macgene, Beijing, China) and cultured in fresh medium. At the 72 hr of transduction, infected 293T cells were transfected with plasmids using PEI. The fluorescence intensities of GFP and mCherry were examined using the Spark 10M Multiscan Spectrum (TECAN, Switzerland).

293T cells were transfected with WT, T17M, *RHO*-5m-pEGFPN1, or *RHO*-variants-pEGFPN1 plasmids using PEI. After 24 hr or 72 hr of transfection, GFP was observed using a Nikon fluorescence microscope (Nikon, Japan). After 1 week of transfection, the cells were fixed in 100% methanol (chilled at –20°C) at room temperature (RT) for 5 min, incubating the samples were incubated for 10 min with PBS containing 0.1% Triton X-100. Cells were incubated with 5% normal donkey serum (017-000-121, Jackson Laboratory, USA) in PBST for 30 min at RT to block nonspecific binding of the antibodies. The primary antibody used was monoclonal mouse monoclonal anti-rhodopsin (4D2) antibody (1:1000 dilution, ab98887, Abcam, Cambridge, MA). The secondary antibody used was donkey anti-mouse antibody conjugated with Alexa Fluor 568 (1:800 dilution, A10037, Invitrogen). Transfected cells were counterstained with DAPI (1:5000 dilution, C0060, Solarbio, Beijing, China) and examined under a Nikon *n-Storm* confocal microscope (Nikon, Japan).

The iPSCs were tested negative for mycoplasma contamination. On the day of transfection, fresh R1 medium containing 10 μM Y-27632 (ROCK inhibitor, STEMCELL, Canada) was added to the iPSCs. iPSCs from the healthy male donor and the male adRP patient were transfected with AAV-EFS-SaCas9-U6-sgRNA-p2a-Puro plasmid using Lipofectamine Stem Reagent (STEM00003, Thermo Fisher Scientific) according to the manufacturer's protocol. For drug selection, R1 medium containing 0.1–0.2 μg/mL puromycin (A1113803, Life, USA ~15 days) was added after 24 hr of transfection.

## Subretinal injection

*RHO* humanized mice were treated on the postnatal month 1.5–1.8 (P1.5–1.8 m) with AAV vectors of AAV-EGFP or a 1:1 mixture of AAV-EGFP and AAV-SaCas9/17-Sg2 or a 1:1 mixture of AAV-EGFP and AAV-SaCas9/CTRL at different doses via subretinal injection in both eyes as described previously (*Hu et al., 2020*; *Wu et al., 2015*). A summary of mouse experiments was provided in *Supplementary file 1b*. Eyes with severe hemorrhage or leakage of the vector solution from the subretinal space into the vitreous were excluded from further study. AAV-SaCas9/17-Sg2 were also delivered to bilateral subretinal space of NHP as described previously (*Maeder et al., 2019*).

## Cell and tissue collection

At 3 mpi, two retinas from each mouse were dissected under a Nikon fluorescence microscope (Nikon, Japan). GFP+ and GFP- areas of the treated retinas were cut for further analysis, including gDNA, RNA, and protein isolation. At 92 dpi, NHP were sacrificed and their retinas were obtained. The gDNA was extracted from retinas or cells and isolated using a DNA isolation kit (DC102, Vazyme, Nanjing,

China). RNA was extracted and isolated from retinas or cells using RNA isolation (RC101, Vazyme, Nanjing, China). In addition, gDNA and RNA were also extracted using TRIzol (15596026, Invitrogen, USA).

## On-target and off-target analysis

The genomic regions flanking the sgRNA target sites were amplified by PCR using 50–100 ng of gDNA as a template. Primers used were listed in *Supplementary file 1i*. For T7EI analysis, 500 ng of purified PCR products (D2500, Gel Extraction Kit, OMEGA, USA) were denatured and reannealed in NEB buffer 2 (M0302S, NEB, USA): 95°C, 2 min; 95–85°C at –2°C/s; 85–25°C at –0.1°C/s. Upon reannealing, 10 U of T7EI (M0302S, NEB, USA) was added and incubated at 37°C for 30 min. Afterward, the product was analyzed on 2.5% agarose gels using a Gel Doc imaging system (Bio-Rad, USA). The amplicons obtained previously were also subjected to clone sequencing for identification of on-target indels. This consisted of cloning the PCR products using the ZTOPO BLUNT/TA with the Zero Background cloning kit following the manufacturer's instructions (ZC211-2, ZOMANBIO, Beijing, China). In addition, PCR products for targeted deep sequencing were prepared as described in the Hi-TOM Kit19 (Novogene, Beijing, China). Meanwhile, mixed samples were sequenced on the Illumina HiSeq platform (https://www.hi-tom.net/hi-tom/documentation.html) (*Hu et al., 2020*; *Liu et al., 2019*).

For off-target analysis, 293T cells were transfected with sgRNA plasmid or CTRL plasmid using PEI. One week after transfection, gDNA was extracted. Off-target sites were obtained from the Benchling's CRISPR Tool. The genomic regions that were flanking off-target sites were amplified by PCR and then validated using the T7E1 assay and Sanger sequencing. In addition, off-target sites were identified using the Cas-OFFinder online tool (http://www.rgenome.net/cas-offinder) (*Bae et al., 2014*). PCR amplicons were analyzed by NGS using the Illumina NovaSeq platform (Illumina, San Diego, CA, USA). Furthermore, gDNA of 293T cells with or without transfection was tested by WGS using the Illumina NovaSeq platform (Illumina, San Diego, CA, USA).

## RNA analysis

Five hundred nanograms of RNA was used for cDNA synthesis using the One-Step gDNA Removal and cDNA Synthesis Supermix Kit (AT311, TransGen Biotech) at a final volume of 20 μL and cDNA was then diluted by adding 80 μL of RNase-free $H_2O$. qPCRs were performed using the ABI7500 Real-Time PCR Detection System (Carlsbad, CA, USA). Each 20 μL reaction contains 10 μL of TransStart Top Green qPCR SuperMix (AQ132, TransGen Biotech), 0.2 μM of each forward and reverse primer, and 1 μL diluted cDNA. All reactions were performed in triplicates. The oligonucleotide sequences are listed in *Supplementary file 1i*.

## WB analysis

Cells or mouse retinas were homogenized in 200 μL of RIPA buffer (C1053+, Applygen, Beijing, China, plus protease inhibitors). Protein quantification was performed using a bicinchoninic acid kit (P1513, Applygen, Beijing, China). Ten micrograms of protein extract loaded onto a 10% sodium dodecyl sulfate-polyacrylamide gel. Following electrophoresis, proteins were transferred onto a PVDF membrane. The membrane was blocked with 5% milk in TBST (B1009, Applygen, Beijing, China) and incubated with a primary mouse monoclonal 1D4 (1:1000 dilution, ab5417, Abcam, Cambridge, MA), rabbit anti-CRISPR-Cas9 antibody (1:5000 dilution, ab203943, Abcam, Cambridge, MA), mouse monoclonal anti-β-actin antibody (1:5000 dilution, AF0003, Beyotime, Shanghai, China), or mouse monoclonal anti-GAPDH antibody (1:5000 dilution, ZL9002, ProteinTech Group, Chicago, IL, USA) in primary antibody diluent for western (C1240, Applygen, Beijing, China) overnight at 4°C, respectively. The next day, after washing three times with 1× TBST (B1109, Applygen, Beijing, China), the membrane was incubated with horse radish peroxidase-conjugated goat anti-mouse IgG antibody (A0216, Beyotime, Shanghai, China) or goat anti-rabbit IgG antibody (A0208, Beyotime, Shanghai, China) at 1:1000 dilution for 1 hr at RT. GAPDH or β-actin was used as a loading control. Blots were analyzed using a ChemiDoc MP imaging system (Tanon Science & Technology Co., Shanghai, China).

## Histological analysis and morphological measurements

For cryosectioning, eyes were enucleated and fixed with 4% paraformaldehyde in PBS or FAS eyeball fixative solution (G1109, Servicebio, Beijing, China), the lens and cornea were removed, and the

eye cups were embedded in optical cutting temperature compound (OCT, 4583, Tissue-Tek, Sakura Finetek, Torrance, CA) and frozen in liquid nitrogen. The eye cup was sectioned to a thickness of 7 μm using a cryostat. For histological analysis, sections were collected at regular intervals from approximately 24 sites per eye. Sections were stored at –80°C and used within 2–3 days.

Before use, the cryosections were fixed on slides for 10 min with acetone. The sections were then washed thrice with PBS, followed by IF staining. The primary antibodies used were monoclonal mouse 4D2 antibody and biotinylated peanut agglutinin (PNA; 1:200 dilution, B-1075, Vector, CA, USA). The secondary antibodies used were donkey anti-mouse conjugated with Alexa Fluor 568 for 4D2, and rhodamine (TRITC)-conjugated streptavidin (1:200 dilution, 123126, Jackson ImmunoResearch Laboratories, West Grove, PA, USA) for PNA. 4D2 and PNA were used to identify the rod and cone photoreceptors, respectively. Cryosections were also stained with HE, and whole slide images were obtained using a NanoZoomer digital slide scanner (Hamamatsu Photonics, Hamamatsu City, Japan).

## Full-field ERG and mfERG

The ERG responses of mice (n≥3) were recorded using the Espion $E^2$ recording system (Diagnosis LLC, Lowell, MA, USA). As shown in *Supplementary file 1b* and our previous study (*Liu et al., 2022*), the ERG recordings were initially conducted at 3 mpi under dark- (scotopic) and light-adapted (photopic) conditions. To record an mfERG for NHP, a confocal scanning laser ophthalmoscope is connected to an mfERG device (RETIscan, Roland Consult, Wiesbaden, Germany) according to the manufacturer's instructions.

## OCT analysis for NHP

All NHPs were sedated, situated, and imaged using a Spectralis HRA + OCT (Heidelberg Engineering, Inc, Germany) scanning laser ophthalmoscope according to the manufacturer's instructions. The macula and subretinal injection area were covered.

## Quantification and statistical analysis

The results were evaluated using two-tailed unpaired Student's t-test and are presented as the mean ± SEM. GraphPad Prism 9 (GraphPad Software, La Jolla, CA, USA) was used for statistical analyses. Statistical significance was set at $p < 0.05$.

## Acknowledgements

The author thanks Dr. Juan Du for the gift of the AAV-EFS-SaCas9-p2a-Puro plasmid and thanks for Prof. Xining Zhao for his helpful insight in revising this article. This article was supported by the National Natural Science Foundation of China (grant nos.: 81770966) and the Beijing Natural Science Foundation of China (grant nos.: 19JCZDJC64000(Z)). The sponsors and/or funding organization had no role in the design or conduct of this research.

## Additional information

### Competing interests

Jing Qiao, Fan Zhang: is an employee of Beijing Chinagene Co.,LTD and was employed by Beijing Chinagene Co.,LTD at the time this work was conducted. The author has no other relevant affiliations or financial involvement with any organization or entity with a financial interest in or financial conflict with the subject matter or materials discussed in the manuscript apart from those disclosed. The other authors declare that no competing interests exist.

### Funding

| Funder | Grant reference number | Author |
|--------|------------------------|--------|
| the National Natural Science Foundation of China | 81770966 | Liping Yang |

| Funder | Grant reference number | Author |
|---|---|---|
| the Beijing Natural Science Foundation of China | 19JCZDJC64000(Z) | Liping Yang |

The funders had no role in study design, data collection and interpretation, or the decision to submit the work for publication.

## Author contributions

Xiaozhen Liu, Conceptualization, Resources, Data curation, Software, Formal analysis, Validation, Investigation, Methodology, Writing – original draft, Project administration, Writing - review and editing; Jing Qiao, Resources, Software, Formal analysis, Validation, Investigation, Writing – original draft, Project administration; Ruixuan Jia, Data curation, Software, Formal analysis, Validation, Investigation, Methodology; Fan Zhang, Software, Formal analysis, Investigation, Methodology; Xiang Meng, Validation, Investigation, Methodology; Yang Li, Data curation, Investigation; Liping Yang, Conceptualization, Data curation, Formal analysis, Funding acquisition, Investigation, Methodology, Writing – original draft, Project administration, Writing - review and editing

## Author ORCIDs

Xiaozhen Liu ⬛ http://orcid.org/0000-0001-9893-1986
Liping Yang ⬛ http://orcid.org/0000-0002-4239-228X

## Ethics

Human subjects: The study was approved by the Medical Scientific Research Ethics Committee of Peking University Third Hospital (Research License 2021262). The procedures were performed in accordance with the tenets set forth in the Declaration of Helsinki. All patients provided written informed consent for this study.

Approval was obtained from the Peking University Health Science Center Ethics Committee for Experimental Animal Research (Research License LA20200473). All procedures were performed according to the regulations of the Association for Research in Vision and Ophthalmology's statement for the use of animals in ophthalmic and vision research. All mice were maintained in accordance with the guidelines of the Association for the Assessment and Accreditation of Laboratory Animal Care.Four adult, female NHP were bred and maintained at JOINN Laboratories (Suzhou, China), approval was obtained from the JOINN Laboratories Ethics Committee for Experimental Animal Research (Research License ACU21-1108).

## Decision letter and Author response

Decision letter https://doi.org/10.7554/eLife.84065.sa1
Author response https://doi.org/10.7554/eLife.84065.sa2

# Additional files

## Supplementary files

• Supplementary file 1. Supplementary tables. (a) Results of TA cloning and Sanger sequencing in 293T cells. (b) A comprehensive summary of mouse experiments. (c) Results of TA cloning and Sanger sequencing in AAV-based SaCas9/17-Sg2-treated Mut-*Rho*$^{wt/hum}$ retinas. (d) Off-target sites of 17-Sg2 obtained from Benchling (https://www.benchling.com/). (e) Off-target sites of 17-Sg2 obtained from Cas-OFFinder (http://www.rgenome.net/cas-offinder) and NGS results. (f) Off-target sites of 17-Sg2 obtained from Cas-OFFinder (http://www.rgenome.net/cas-offinder) and NGS results. (g) The number of off-target sites of 17-Sg2 obtained from Cas-OFFinder (http://www.rgenome.net/cas-offinder). (h) Whole-genome sequencing results of off-target activity for SaCas9/17Sg-2. (i) List of primers.

• MDAR checklist

• Source data 1. Source data files of the raw unedited gels or blots.

• Source data 2. Source data files of the gels or blots with the relevant bands clearly labelled.

• Source data 3. Source data files of NGS results and ERG statistical data.

## Data availability

All data generated or analysed during this study are included in the manuscript and supporting file; Source Data files have been provided in Source Data 1, 2 and 3.

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
