## [Editor Report]

This work provides a valuable allele-specific gene editing therapeutic approach to selectively target the human RHO-T17M mutation, one of the most frequent genetic causes of autosomal dominant retinitis pigmentosa patients. Overall, the data is solid.

---

## [Decision Letter]

**Decision letter after peer review:**

Thank you for submitting your article "Allele-specific gene editing approach for vision loss restoration in RHO-associated Retinitis Pigmentosa" for consideration by *eLife*. Your article has been reviewed by 2 peer reviewers, and the evaluation has been overseen by a Reviewing Editor and Lois Smith as the Senior Editor. The reviewers have opted to remain anonymous.

Essential revisions:

There are several concerns to be addressed:

1. Detailed description of the discovery phase of the sgRNAs.

2. Knockdown efficiency in photoreceptors at the cellular level in vivo.

3. Clarification of the choice of mouse mutants (RHO humanized mice instead of Rho-T17M mice).

4. Inclusion of two more controls (wild-type mice and untreated mutant mice).

5. Examination of other retinal cell types.

6. Evaluation of the toxicity of the vectors on mouse retinas.

*Reviewer #1 (Recommendations for the authors):*

Some additional in vivo experiments would greatly strengthen the study:

1. Show the pathology and progression rate of rod death in Rho-T17M mice (not Rho-5m) as a baseline.

2. Test the gene editing efficiency of these vectors in rods at the cellular level using T17M mice (not entire retina or non-specific retinal cells).

3. Check if the vectors rescue vision in T17M mice vision and determine the duration of improvement.

4. Perform histology on mouse retina that is treated with these vector to check for any toxicity.

*Reviewer #2 (Recommendations for the authors):*

Lines 81-82: RHO-T17M is uncommon in Chinese patients.

Line 183: Figure 3M is cited in the main text (line 183) but is absent in the Figure.

The legend of Figure 7: It lacks descriptions for panel C and D.

Lines 144-146, WB results showed that the expression of Rho in HEK293T cell lines transfected with 17-sg1 or sg2 plasmid was strongly reduced compared with RHOwt cells. Please provide the statistical analysis of the WB result.

Lines 178-180: The results of Hi-Tom sequence of SaCas9 and 17-Sg2-treated P1 iPSCs and P1 iPSCs colony were different. Please explain why the ipS and iPS clones in this study produce different indels. And state how the colonies from P1 iPS can further confirm the allele-specific cutting.

[Editors' note: further revisions were suggested prior to acceptance, as described below.]

Thank you for resubmitting your work entitled "Allele-specific gene editing approach for vision loss restoration in RHO-associated Retinitis Pigmentosa" for further consideration by *eLife*. Your revised article has been evaluated by Lois Smith (Senior Editor) and Reviewing Editors.

The manuscript has been improved but there are some remaining issues that need to be addressed, as outlined below:

The current study identified SaCas9 guide RNA that is highly active and specific to human T17M RHO allele, evidenced in HEK293T cells and iPSCs in vitro, as well as RHO humanized mice. This therapeutic approach was supported with solid in vitro data and showed long-term beneficial effects with improved retinal function and preservation of photoreceptors in mutant humanized heterozygous mice. However, some of the key concerns raised in the previous review need to be addressed.

1. Editing efficiency in photoreceptors at cellular level in vivo. Please check the DNA or RNA of rods in situ, report the percentage of rods which were successfully edited by the vector.

2. Clarification of the choice of mouse mutants (RHO humanized mice instead of Rho-T17M mice). The authors need to include the following information regarding RHO humanized mice in the Introduction session: please change "humanized mice" to "hRho-knock-in". Please include the following statement "the retinal degeneration might be caused by any one of the five variants (T17M, G51D, G114R, R135W and P171R)." Please discuss and refer to the literature if each of the five mutations is pathogenic in heterozygous status. Also please include the following reviewer comment "as the five variants are in cis on one allele, gene editing on T17M alone would erase the effect of other four variants because such editing creates frameshift variants at the N-terminal region before the other four".

3. Please edit the language to tone down any overstatement. For example, in the sentence "Humanized animals are revolutionizing our ability to study and model disease and permit testing of the therapeutic effect of CRISPR/Cas9 in preclinical in vivo systems without putting humans at risk", the authors may consider revising as "Animals with human gene knock-in allow us to study and model disease and permit testing of the therapeutic effect of CRISPR/Cas9 in preclinical in vivo systems". The authors are encouraged to check through and edit the manuscript.

---

## [Author Response]

Essential revisions:There are several concerns to be addressed:1. Detailed description of the discovery phase of the sgRNAs.

Thanks for your suggestion. Presently, there are several websites (CHOPCHOP, Benchling, E-CRISP, etc.) which can be used to design sgRNA easily, herein, sgRNA were designed based on the Benchling website (https://www.benchling.com/). Every sgRNA design website will provide off-target data, these data will help us to evaluate the off-target sites of sgRNA. Herein, Off-target sites were obtained from the Benchling’s CRISPR Tool. For more detailed assessment of off target, off-target sites were identified using the Cas-OFFinder online tool (http://www.rgenome.net/cas-offinder), the detailed method is listed in ref. 1. This was presented in the Material and Methods on Page 31 Line 623 to 627.

ref. 1: Bae S, Park J, Kim JS. Cas-OFFinder: a fast and versatile algorithm that searches for potential off-target sites of Cas9 RNA-guided endonucleases. Bioinformatics. 2014 May 15;30(10):1473-5. doi: 10.1093/bioinformatics/btu048.

2. Knockdown efficiency in photoreceptors at the cellular level *in vivo.*

Thanks for your suggestion. Thanks to the fact that *RHO* gene expressed in rods specially in retina, the mutant human rhodopsin mRNA levels could be evaluated using sequencing. The aim of allele-specific gene-editing treatment was to attenuate or ablate the expression of the mutant allele. Herein, we then determined whether the indels created by SaCas9/17-Sg2 at the DNA level could result in decreased mRNA levels of the *RHO*-5m allele. AAV-EGFP and AAV-SaCas9/17-Sg2 (1:1 mixture) were co-delivered into both eyes of the *Rho^hum/m-hum^* mice. *Rho^hum/m-hum^* mouse, a humanized mouse model carried a human *RHO*-WT allele and *RHO*-5m allele. Truncated cleaved products treated with T7E1 nuclease were observed in treated *Rho^hum/m-hum^* retinas (Figure 7). We also employed Hi-Tom sequencing, Sanger sequencing of RT-PCR products to compare the amount of mRNA transcripts of *RHO*-WT and *RHO*-5m in *Rho^hum/m-hum^* retinas at 3 mpi. The results indicated that the percentage of WT transcripts was statistically higher in treated retinas than that in untreated retinas (67.47 % vs. 48.51% (*p*<0.005) and 60.32 % vs. 42.66% (*p*<0.05), respectively), and the mutant transcript was statistically lower in treated retinas than that in untreated retinas (32.53 % vs. 51.49% (*p*<0.005) and 39.68 % vs. 57.34% (*p*<0.05), respectively, Figure 7C-7D), indicating the mRNA expression of *RHO*-5m decreased by 36.82% after treatment (Hi-Tom sequencing). This was presented in the Results on Page 16 Line 315 to 324.

3. Clarification of the choice of mouse mutants (RHO humanized mice instead of Rho-T17M mice).

Thanks for your suggestion.

1) Mutant *RHO* proteins cause disease via either a dominant negative or a toxic gain-of-function effect. The aim of this study is to evaluate whether can effectively treat human *RHO*-adRP by gene-editing knockout of the mutant protein. Humanized mice permitted permit testing of the therapeutic effect of CRISPR/Cas9 in preclinical *in vivo* systems, without putting humans at risk. See Figure 5A and Figure 5—figure supplement 1C, there was 8 mismatches between 17-Sg2 and the corresponding mouse *Rho* sequence, if we want to design sgRNA to target human mutant *RHO* gDNA, humanized mice were needed.

2) Available *RHO* humanized models are few compared to the number of known *RHO* mutations (more than 200), but it is time-consuming and costly to build humanized mice for each mutation. We want to develop a humanized mouse model having several mutations simultaneously to investigate the therapeutic effect of CRISPR/Cas9 for *RHO*-mediated adRP in preclinical *in vivo* systems. These five mutations in the humanized mouse model occur in the same human *RHO* allele, our study indicated the majority of indels created by the activity of SaCas9/17-Sg2 around T17M DNA sequence in Mut-*Rho^wt/hum^* mice could lead to attenuation or ablation of the *RHO*-5m allele. Thereby, before the gene-editing treatment experiments, we custom designed and generated the humanized mouse models.

3) Our previous study (ref. 2) demonstrated that the pattern of retinal degeneration caused by mutant human rhodopsin was a typical rod-cone decay, thereby Mut-*Rho^wt/hum^* mice were affected with adRP, mimicking human adRP phenotypes. In conclusion, at DNA level, this mouse model could be used to analyze gene-editing efficiency, at RNA level, it could be used to analyze whether the mutant human rhodopsin was knocked down, and it could be employed to evaluate whether its adRP phenotypes were rescued after treatment.

ref. 2: Liu X, Jia R, Meng X, Li Y, Yang L. Retinal degeneration in humanized mice expressing mutant rhodopsin under the control of the endogenous murine promoter. Exp Eye Res. 2022 Feb;215: 108893. doi: 10.1016/j.exer.2021.108893.

4. Inclusion of two more controls (wild-type mice and untreated mutant mice).

Thanks for your suggestion. We added these two control groups, see Figure 8—figure supplement 1. We added these sentences “Besides, the retinal functions of AAV-SaCas9/17-Sg2-treated Mut-*Rho^wt/hum^* mice were better than that of untreated age-matched Mut-*Rho^wt/hum^* mice, and had no significant difference when compared to that of untreated same-age WT mice (Figure 8—figure supplement 1B) (the Results on Page 17 Line 346 to 349).” and “ at 3 mpi, the retinal functions of untreated WT and Mut-*Rho^wt/hum^* mice were significantly better than that of AAV-SaCas9/17-Sg2- and AAV-SaCas9/CTRL -treated mice, while the remarkable long-term therapeutic effect in AAV-SaCas9/17-Sg2- treated mice was detected, implying that: (a) compared to NHP and human eyes, the eye volume of mouse was relatively small, subretinal injection which was an invasive operation would result in retinal detachment, certainly leading to a damage to retinal functions. However, as times went on, retinal detachment would recover, exerting less effect on the retinal functions, indicating that the long-term therapeutic effect of drug should be observed. (b) In preclinical experiments or even clinical trials, one variable between treated group and control group (gene-editing drug with 17-Sg2 or not) was essential, to ensure the statistical results was reliable. For instance, between untreated Mut-*Rho^wt/hum^* mice and treated ones, there were two variables: (a) subretinal injection, which is an invasive operation, (b) the gene-editing drug including AAV-SaCas9/17-Sg2 or not (the Discussion on Page 27-28 Line 555 to 568).

5. Examination of other retinal cell types.

Thanks for your suggestion. We added Figure 9—figure supplement 3 to Supplementary file. Protein kinase C (PKC-α) for rod bipolar cells, calbindin for horizontal cells, and CRALBP for RPE and Müller cells which span throughout the retina vertically, compared to untreated mice and AAV-SaCas9/CTRL-treated mice at 9 mpi and 11 mpi, more and healthier bipolar cells, horizontal cells and Müller cells were detected in AAV-SaCas9/17-Sg2-treated retinas. This was added to the Results on Page 18 Line 361 to 367.

6. Evaluation of the toxicity of the vectors on mouse retinas.

Thanks for your suggestion. First, the first factor that may raise safety concerns for allele-specific CRISPR/Cas9 drug development is the specificity of SaCas9/SgRNA (ref. 3). This study demonstrated that SaCas9/17-Sg2 did not disrupt human WT *RHO* both *in vitro* and *in vivo* (Figure 5-figure supplement 1) and no off-target effect (Figure 10). Second, up to 11 mpi, no detectable toxic effects in WT mouse retinas were identified when comparing the retinal structures of treated and untreated mice (Figure 4D). In addition, AAV-SaCas9/17-Sg2 was also delivered to subretinal space of nonhuman primate (NHP), ERG and HE staining indicated no toxicity of the vector (Figure 10—figure supplement 2).

ref. 3: Maeder ML, Stefanidakis M, Wilson CJ, et.al. Development of a gene-editing approach to restore vision loss in Leber congenital amaurosis type 10. Nat Med. 2019 Feb;25(2):229-233. doi: 10.1038/s41591-018-0327-9.

Reviewer #1 (Recommendations for the authors):Some additional *in vivo* experiments would greatly strengthen the study:1. Show the pathology and progression rate of rod death in Rho-T17M mice (not Rho-5m) as a baseline.

Thanks for your suggestion.

1) Mutant *RHO* proteins cause disease via either a dominant negative or a toxic gain-of-function effect. The aim of this study is to evaluate whether can effectively treat human *RHO*-adRP by gene-editing knockout of the mutant protein. Humanized mice permitted permit testing of the therapeutic effect of CRISPR/Cas9 in preclinical *in vivo* systems, without putting humans at risk. See Figure 5A and Figure 5—figure supplement 1C, there was 8 mismatches between 17-Sg2 and the corresponding mouse *Rho* sequence, if we want to design sgRNA to target human mutant *RHO* gDNA, humanized mice were needed.

2) Available *RHO* humanized models are few compared to the number of known *RHO* mutations (more than 200), but it is time-consuming and costly to build humanized mice for each mutation. We want to develop a humanized mouse model having several mutations simultaneously to investigate the therapeutic effect of CRISPR/Cas9 for *RHO*-mediated adRP in preclinical in vivo systems. These five mutations in the humanized mouse model occur in the same h*RHO* allele, our study indicated the majority of indels created by the activity of SaCas9/17-Sg2 around T17M DNA sequence in Mut-*Rho^wt/hum^* mice could lead to attenuation or ablation of the *RHO*-5m allele. Thereby, before the gene-editing treatment experiments, we custom designed and generated the humanized mouse models.

3) Our previous study (ref. 2) demonstrated that the pattern of retinal degeneration caused by mutant human rhodopsin was a typical rod-cone decay, thereby Mut-*Rho^wt/hum^* mice were affected with adRP, mimicking human adRP phenotypes. In conclusion, at DNA level, this mouse model could be used to analyze gene-editing efficiency, at RNA level, it could be used to analyze whether the mutant human rhodopsin was knocked down, and it could be employed to evaluate whether its adRP phenotypes were rescued after treatment.

ref. 2: Liu X, Jia R, Meng X, Li Y, Yang L. Retinal degeneration in humanized mice expressing mutant rhodopsin under the control of the endogenous murine promoter. Exp Eye Res. 2022 Feb;215: 108893. doi: 10.1016/j.exer.2021.108893.

2. Test the gene editing efficiency of these vectors in rods at the cellular level using T17M mice (not entire retina or non-specific retinal cells).

Thanks for your suggestion. Thanks to the fact that *RHO* gene expressed rods specially in retina, the mutant human rhodopsin mRNA levels could be evaluated using sequencing. The aim of allele-specific gene-editing treatment was to attenuate or ablate the expression of the mutant allele. Herein, we then determined whether the indels created by SaCas9/17-Sg2 at the DNA level could result in decreased mRNA levels of the *RHO*-5m allele. AAV-EGFP and AAV-SaCas9/17-Sg2 (1:1 mixture) were co-delivered into both eyes of the *Rho^hum/m-hum^* mice. *Rho^hum/m-hum^* mouse, a humanized mouse model carried a human *RHO*-WT allele and *RHO*-5m allele. Truncated cleaved products treated with T7E1 nuclease were observed in treated *Rho^hum/m-hum^* retinas (Figure 7). We employed Hi-Tom sequencing, Sanger sequencing of RT-PCR products to compare the amount of mRNA transcripts of *RHO*-WT and *RHO*-5m in *Rho^hum/m-hum^* retinas at 3 mpi. The results indicated that the percentage of WT transcripts was statistically higher in treated retinas than that in untreated retinas (67.47 % vs. 48.51% (*p*<0.005) and 60.32 % vs. 42.66% (*p*<0.05), respectively), and the mutant transcript was statistically lower in treated retinas than that in untreated retinas (32.53 % vs. 51.49% (*p*<0.005) and 39.68 % vs. 57.34% (*p*<0.05), respectively, Figure 7C-7D), indicating the mRNA expression of *RHO*-5m decreased by 36.82% after treatment (Hi-Tom sequencing). This was presented in the Results on Page 16 Line 315 to 324.

3. Check if the vectors rescue vision in T17M mice vision and determine the duration of improvement.

Thanks for your suggestion. Mutant *RHO* proteins cause disease via either a dominant negative or a toxic gain-of-function effect. The aim of this study is to evaluate whether can effectively treat human *RHO*-adRP by gene-editing knockout of the mutant protein. These five mutations in the humanized mouse model occur in the same human *RHO* allele, once the mutant allele around the T17M DNA sequence was cut by Cas9/sgRNA, the downstream sequence of the cut site would be abolished due the DNA NHEJ repair. Our results herein confirmed the above theory (Figure 6-7). That’s no doubt that our humanized models could be used gene-editing efficiency at DNA level. In addition, the previous study (ref. 5) has established a transgenic mouse that expresses both normal and mutant murine rhodopsin in rods, the mutant rhodopsin contains 3 amino acids substitutions (V20G, P23H and P27L) simultaneously and results in a slowly progressing rods and cones degeneration. The structural and functional defects of P23H mice closely mimic those in patients carrying the same mutation. Several previous studies (ref. 6-7) indicated that rhodopsin with T17M, P23H, etc. could result in late onset adRP in transgenic mouse models. Our previous study (ref. 2) demonstrated that the pattern of retinal degeneration caused by mutant human rhodopsin was a typical rod-cone decay, thereby Mut-*Rho^wt/hum^* mice were affected with adRP, mimicking human adRP phenotypes. In conclusion, Mut-*Rho^wt/hum^* mice can be used to evaluate the gene-editing efficiency, retinal functions improvement and photoreceptor preservation after gene-editing treatment. Administration of this therapeutic drug resulted in a long-term (up to 11 months after treatment) improvement of retinal function and preservation of photoreceptors in the mutant humanized heterozygous mice. Our study demonstrated a dose-dependent therapeutic effect in vivo.

ref. 2: Liu X, Jia R, Meng X, Li Y, Yang L. Retinal degeneration in humanized mice expressing mutant rhodopsin under the control of the endogenous murine promoter. Exp Eye Res. 2022 Feb; 215:108893. doi: 10.1016/j.exer.2021.108893.

ref. 5: Naash MI, Hollyfield JG, al-Ubaidi MR, Baehr W. Simulation of human autosomal dominant retinitis pigmentosa in transgenic mice expressing a mutated murine opsin gene. Proc Natl Acad Sci U S A. 1993 Jun 15;90(12):5499-503. doi: 10.1073/pnas.90.12.5499.

ref. 6: Olsson, J. E., Gordon, 1. W., Pawlyk, B. S., Roof, D., Hayes, A., Molday, R. S., Mukai, S., Cowley, G. S., Berson, E. L., and Dryja, T. P., 1992, Transgenic mice with a rhodopsin mutation (Pr023His): a mouse model of autosomal dominant retinitis pigmentosa, Neuron 9:815-830.

ref. 7: Kunte MM, Choudhury S, Manheim JF, Shinde VM, Miura M, Chiodo VA, Hauswirth WW, Gorbatyuk OS, Gorbatyuk MS. ER stress is involved in T17M rhodopsin-induced retinal degeneration. Invest Ophthalmol Vis Sci. 2012 Jun 20;53(7):3792-800. doi: 10.1167/iovs.11-9235.

4. Perform histology on mouse retina that is treated with these vector to check for any toxicity.

Thanks for your suggestion. First, the first factor that may raise safety concerns for allele-specific CRISPR/Cas9 drug development is the specificity of SaCas9/SgRNA (ref. 3). This study demonstrated that SaCas9/17-Sg2 did not disrupt human WT RHO both *in vitro* and *in vivo* (Figure 5—figure supplement 1) and no off-target effect (Figure 10). Second, up to 11 mpi, no detectable toxic effects in WT mouse retinas were identified when comparing the retinal structures of treated and untreated mice (Figure 4D). In addition, AAV-SaCas9/17-Sg2 was also delivered to subretinal space of nonhuman primate (NHP), ERG and HE staining indicated no toxicity of the vector (Figure 10—figure supplement 2).

ref. 3: Maeder ML, Stefanidakis M, Wilson CJ, et.al. Development of a gene-editing approach to restore vision loss in Leber congenital amaurosis type 10. Nat Med. 2019 Feb;25(2):229-233. doi: 10.1038/s41591-018-0327-9.

Reviewer #2 (Recommendations for the authors):Lines 81-82: RHO-T17M is uncommon in Chinese patients.

Thanks for your suggestion. In our previous study and clinical practice (Peking university third hospital and Beijing Tongren hosptial), several adRP families with many affected individuals had *RHO*-T17M mutation, and in HGMD (http://www.hgmd.cf.ac.uk/ac/index.php) database, at least 7 references reported this mutation, thereby, we indicated “some Chinese adRP patients carried *RHO*-T17M mutation”.

ref. 4: Liu X, Tao T, Zhao L, Li G, Yang L. Molecular diagnosis based on comprehensive genetic testing in 800 Chinese families with non-syndromic inherited retinal dystrophies. Clin Exp Ophthalmol. 2021 Jan;49(1):46-59. doi: 10.1111/ceo.13875.

Line 183: Figure 3M is cited in the main text (line 183) but is absent in the Figure.

Thanks for your suggestion. We have corrected this mistake, Figure 3M should be Figure 3L.

The legend of Figure 7: It lacks descriptions for panel C and D.

Thanks for your suggestion. We have corrected this mistake. This was presented in the Figure Legends on Page 59 Line 1219 to 1221.

Lines 144-146, WB results showed that the expression of RHO in HEK293T cell lines transfected with 17-sg1 or sg2 plasmid was strongly reduced compared with RHOwt cells. Please provide the statistical analysis of the WB result.

Thanks for your suggestion. We added the statistical analysis of the WB result to Figure 2, as illustrated in Figure 2E, densitometric analysis of immunoblots performed on RHOwt and RHO17 cells transfected with 17-Sg1 and Sg2 plasmid, respectively. This was presented in the Figure Legends on Page 56 Line 1156 to 1159.

Lines 178-180: The results of Hi-Tom sequence of SaCas9 and 17-Sg2-treated P1 iPSCs and P1 iPSCs colony were different. Please explain why the ipS and iPS clones in this study produce different indels. And state how the colonies from P1 iPS can further confirm the allele-specific cutting.

Thanks for your suggestion. In this study, iPSCs were obtained from UCs using the integration-free CytoTune 2.0 Sendai Reprogramming Kit (A16517, Thermo Fisher Scientific, MA, USA), the detailed protocol was provided by the manufacturer. Herein, sequencing results of treated P1 iPSCs was listed herein, P1 iPSCs colony was picked from P1 iPSCs, which were better shaped, well-conditioned cells, in order to validate the result and to obtain more objective result, experiment of P1 iPSCs colony was performed repeatedly, we just presented the objective results of DNA sequencing, just like qPCR results, if we want to get credible results, the experiment will be performed in triplicate.

[Editors' note: further revisions were suggested prior to acceptance, as described below.]

The manuscript has been improved but there are some remaining issues that need to be addressed, as outlined below:The current study identified SaCas9 guide RNA that is highly active and specific to human T17M RHO allele, evidenced in HEK293T cells and iPSCs in vitro, as well as RHO humanized mice. This therapeutic approach was supported with solid in vitro data and showed long-term beneficial effects with improved retinal function and preservation of photoreceptors in mutant humanized heterozygous mice. However, some of the key concerns raised in the previous review need to be addressed.1. Editing efficiency in photoreceptors at cellular level in vivo. Please check the DNA or RNA of rods in situ, report the percentage of rods which were successfully edited by the vector.

Thanks for your suggestion.

As we know, CRISPR/Cas9 system can lead to permanent DNA modification because of endonuclease properties of Cas9, thereby it is essential to calculate the gene-editing efficiency at DNA level. While the aim of CRISPR/Cas9 gene-editing is to alter gene expression transcriptionally and translationally. Mutant *RHO* proteins cause disease via either a dominant negative or a toxic gain-of-function effect. So, it is ideal and promising to treat human *RHO*-adRP by gene-editing knockout of the mutant protein, thereby to increase WT and mutant *RHO* protein ratio. That is the core idea of this study.

1) Initially, to test the editing efficiency of rods at the cellular level *in vivo*, we tried to use Fluorescence activated Cell Sorting (FACS) technique to sort AAV-infected rods by labeling the cells with specific marker, then to calculate the gene editing efficiency. However, in practice, we have encountered a lot of problems, for instance, during FACS experiments, it was difficult to a) digest the entire retinal tissue into single cells, b) prevent excessive rods loss, and we could not obtain abundant gDNA from few sorted cells. Previously, several articles reported that under a fluorescent dissection microscope, GFP+ retinal areas which represented AAV-infected areas could be obtained (*ref. 1~2*). Herein, AAV-EGFP and AAV-SaCas9/17-Sg2 (1:1 mixture) were co-delivered into both eyes of the humanized mouse models. IF analysis indicated that GFP was expressed in the OS/IS and ONL of photoreceptor cells (Figure 4D, Figure 9A, Figure 9—figure supplement 1 and Figure 9—figure supplement 4), showing that SaCas9 and 17-Sg2 were also expressed in OS/IS and ONL, therefore, the gDNA of rods and cones in ONL would be edited.

Neural retina consists of nine layers as illustrated in ref.3. In this study, AAV infected in OS/IS and ONL mainly. Under a fluorescent dissection microscope, we flattened the neural retina (Figure 4C), the GFP+ areas were dissected and gene-editing efficiency in these areas were focused. While in fact, we collected all types of cells apart from rods and cones, including those cells without infection in other retinal layers. So, compared to the gene-editing efficiency in all types of cells (we calculated in this study), actual gene-editing efficiency in rods was expected to be higher.

2) As we said above, the aim of CRISPR/Cas9 gene-editing is to alter gene expression transcriptionally and translationally. To treat *RHO*-adRP, it is promising and important to increase the WT RHO mRNA level and decrease the mutant *RHO* mRNA level. Thanks to the fact that *RHO* gene only expressed in rods both transcriptionally and translationally, its encoding protein rhodopsin is the marker of rods and embedded at high density in the membrane of the rods outer segment (ROS) disks. At DNA level, gene-editing could occur in all types of AAV-infected retinal cells, while at mRNA and protein level, *RHO* expression could be attenuated or ablated only in rods. Fortunately, Hi-Tom sequencing result of RT-PCR products indicated that the mRNA expression of *RHO*-5m decreased by 36.82% after treatment. That is the editing efficiency or in rods (the targeted cell type) in situ at mRNA level in vivo.

3) Our previous study (ref. 4) reported that the mouse *Rho* gene contains 5 exons and shares a high homology at coding sequence (88.3% homology, Supplementary Figure S1) and amino acids sequence (94.8% homology, Supplementary Figure S2) with human *RHO* gene. That is a prerequisite for successful construction of our humanized mice (ref. 4).

Previously, we tried to use fluorescence in situ hybridization (FISH) technique to discriminate mouse and human *RHO* gene, or WT and mutant *RHO* gene after treatment. However, we could not obtain any appropriate probe due to their high homology at coding sequence. Most importantly, if the mutant *RHO* gene was not edited (control groups), the rods would die ultimately, the *RHO* expression would decrease, while the *RHO* expression decreased due to allele-specific gene-editing (SaCas9/17-Sg2-treated-groups, Figure 7C-E). Such decreased expression cannot be detected by FISH or WB, so the difference of gene ablation expression in control and treated groups cannot be counted effectively. Fortunately, Sequencing of RT-PCR products could help us to reach this target. And a previous reference (ref. 1) also used RT-PCR product sequencing to evaluate the mutant and WT RHO allele ratio after allele-specific gene-editing therapy.

4) Presently, commercial primary antibodies for rhodopsin are 4D2 antibody (antibody recognizing N-terminal of rhodopsin protein) and 1D4 antibody (antibody recognizing C-terminal nine amino acids of rhodopsin). However, both cannot discriminate mouse and human rhodopsin due to the high homology at amino acids sequence (94.8% homology). Our previous study (ref. 4) showed that in mutant humanized mice, mutant rhodopsin localized to RIS and cell bodies while WT rhodopsin should correctly be trafficked to ROS. The aim of allele-specific gene-editing treatment was to attenuate or ablate the expression of the mutant *RHO*. “As showed in Figure 9—figure supplement 1A and C, compared to CTRL groups, mutant rhodopsin in RIS and rods bodies (4D2, red) decreased significantly in SaCas9/17-Sg2-treated mouse retinas, indicating the mutant rhodopsin was ablated in rods (the targeted cell type), similar results were also detected at 9 mpi and 11 mpi (Figure 9—figure supplement 3)”. This was presented on the Results on Page 18 Line 364 to 370. The Figure legends was presented on Page 64 to 65 Line 1332 to 1340.

ref. 1: LI, P., KLEINSTIVER, B. P., LEON, M. Y., et al. 2018. Allele-Specific CRISPR-Cas9 Genome Editing of the Single-Base P23H Mutation for Rhodopsin-Associated Dominant Retinitis Pigmentosa. CRISPR J, 1, 55-64.

ref. 2: Benati D, Marigo V, Recchia A. CRISPR/Cas9 Gene Editing in vitro and in Retinal Cells in vivo. Methods Mol Biol. 2019;1834:59-74.

ref. 3: Power M, Das S, Schutze K, et al. Cellular mechanisms of hereditary photoreceptor degeneration – Focus on cGMP [J]. Prog Retin Eye Res, 2020, 74(100772).

ref. 4: Liu X, Jia R, Meng X, Li Y, Yang L. Retinal degeneration in humanized mice expressing mutant rhodopsin under the control of the endogenous murine promoter. Exp Eye Res. 2022 Feb;215: 108893. doi: 10.1016/j.exer.2021.108893.

2. Clarification of the choice of mouse mutants (RHO humanized mice instead of Rho-T17M mice). The authors need to include the following information regarding RHO humanized mice in the Introduction session: please change "humanized mice" to "hRho-knock-in". Please include the following statement "the retinal degeneration might be caused by any one of the five variants (T17M, G51D, G114R, R135W and P171R)." Please discuss and refer to the literature if each of the five mutations is pathogenic in heterozygous status. Also please include the following reviewer comment "as the five variants are in cis on one allele, gene editing on T17M alone would erase the effect of other four variants because such editing creates frameshift variants at the N-terminal region before the other four".

Thanks for your suggestion.

1) In this study, we emphasized the significance of humanized mouse models if we want to develop gene-editing medicine for human RP and explain why we generated and used *RHO*-5m humanized mouse (Introduction and Discussion). Herein, *Rho^wt/hum^* mice, which was heterozygotes with a mouse *Rho*-WT allele and human *RHO*-WT allele, were used to evaluate gene editing specificity. Mut-*Rho^wt/hum^* mice, which were heterozygotes with a mouse *Rho*-WT allele and human *RHO*-5m allele, were used to test cutting efficiency in vivo (Page 11 to 12 on Line 221 to 227). *Rho^hum/m-hum^* mouse, a humanized mouse model carried a human *RHO*-WT allele and *RHO*-5m allele. They were used to test whether *RHO*-5m expression was reduced, and to test whether allele-specific gene editing could increase WT and mutant RHO protein ratio in vivo (Page 15 on Line 304 to 308).

2) We changed “*RHO* humanized mouse model” to “human *RHO*-knock-in mouse model”. And add the sentence “the retinal degeneration might be caused by any one of the five variants (T17M, G51D, G114R, R135W and P171R, *RHO*-5m)”. This was presented on the Introduction on Page 5 Line 103 to 106.

3) The sentence “Previous studies (Mendes et al., 2005, Liu et al., 2021) and data from HGMD have reported t that each of the five heterozygous mutations in *RHO* can cause adRP” was added to the Introduction (Page 26 Line 531 to 532).

4) The sentence “as the five variants are in cis on one allele, allele-specific gene editing on T17M alone would erase the effect of other four variants because such editing creates frameshift variants at the N-terminal region before the other four” was added. The sentence “The existence of the other four variants in the *RHO* coding region would not interrupt the allele specific gene editing for T17M” was deleted. This was presented on the Results on Page 15 Line 297 to 299.

3. Please edit the language to tone down any overstatement. For example, in the sentence "Humanized animals are revolutionizing our ability to study and model disease and permit testing of the therapeutic effect of CRISPR/Cas9 in preclinical in vivo systems without putting humans at risk", the authors may consider revising as "Animals with human gene knock-in allow us to study and model disease and permit testing of the therapeutic effect of CRISPR/Cas9 in preclinical in vivo systems". The authors are encouraged to check through and edit the manuscript.

Thanks for your suggestion. The entire manuscript has also been proofread again to correct any other mistakes and grammatical errors. The sentence “Humanized animals are revolutionizing our ability to study and model disease and permit testing of the therapeutic effect of CRISPR/Cas9 in preclinical in vivo systems without putting humans at risk” was replaced with “Animals with human gene knock-in allow us to study and model disease and permit testing of the therapeutic effect of CRISPR/Cas9 in preclinical in vivo systems”. This was presented on the Discussion on Page 25 Line 520 to 522.